# Marine Cyanobacteria as Sources of Lead Anticancer Compounds: A Review of Families of Metabolites with Cytotoxic, Antiproliferative, and Antineoplastic Effects

**DOI:** 10.3390/molecules27154814

**Published:** 2022-07-27

**Authors:** Benjamín Robles-Bañuelos, Lorena María Durán-Riveroll, Edgar Rangel-López, Hugo Isidro Pérez-López, Leticia González-Maya

**Affiliations:** 1Facultad de Farmacia, Universidad Autónoma del Estado de Morelos, Av. Universidad 1001, Col. Chamilpa, Cuernavaca C.P. 62209, Mexico; benjamin.robles@uaem.edu.mx; 2Laboratorio de Aminoácidos Excitadores/Laboratorio de Neurofarmacología Molecular y Nanotecnología, Instituto Nacional de Neurología y Neurocirugía, Insurgentes Sur 3877, Mexico City 14269, Mexico; raledg@hotmail.com; 3CONACYT—Departamento de Biotecnología Marina, Centro de Investigación Científica y Educación Superior de Ensenada, Baja Cailifornia, Carr. Tijuana-Ensenada 3918, Zona Playitas, Ensenada 22860, Mexico; 4Alfred Wegener Institut-Helmholtz Zentrum Für Polar- Und Meeresforschung, Am Handelshafen 12, 27579 Bremerhaven, Germany; 5Departamento de Biotecnología Marina, Centro de Investigación Científica y Educación Superior de Ensenada, Baja California, Carr. Tijuana-Ensenada 3918, Zona Playitas, Ensenada 22860, Mexico; hugoperez@cicese.mx

**Keywords:** angiogenesis inhibitor, antiproliferative activity, apoptosis, bioactive metabolites, cytoskeleton disruptor, histone deacetylase inhibitor

## Abstract

The marine environment is highly diverse, each living creature fighting to establish and proliferate. Among marine organisms, cyanobacteria are astounding secondary metabolite producers representing a wonderful source of biologically active molecules aimed to communicate, defend from predators, or compete. Studies on these molecules’ origins and activities have been systematic, although much is still to be discovered. Their broad chemical diversity results from integrating peptide and polyketide synthetases and synthases, along with cascades of biosynthetic transformations resulting in new chemical structures. Cyanobacteria are glycolipid, macrolide, peptide, and polyketide producers, and to date, hundreds of these molecules have been isolated and tested. Many of these compounds have demonstrated important bioactivities such as cytotoxicity, antineoplastic, and antiproliferative activity with potential pharmacological uses. Some are currently under clinical investigation. Additionally, conventional chemotherapeutic treatments include drugs with a well-known range of side effects, making anticancer drug research from new sources, such as marine cyanobacteria, necessary. This review is focused on the anticancer bioactivities of metabolites produced by marine cyanobacteria, emphasizing the identification of each variant of the metabolite family, their chemical structures, and the mechanisms of action underlying their biological and pharmacological activities.

## 1. Introduction

Marine ecosystems are reservoirs of biologically active metabolites producers. Many organisms, especially those without physical protection, such as hard shells or spines, have evolved chemical weapons for defense or competition purposes, and some of them are little-known bioactive substances with pharmacological potential as anticancer agents [1]. Among the primary producers in aquatic environments, the functional group “microalga” (which includes cyanobacteria and photosynthetic protists) is the basis of all sea life. These microscopic organisms are responsible for producing, through photosynthesis, biomass and oxygen, among other important primary metabolites, such as carbohydrates, proteins, and lipids.

Currently, more than 1600 molecules from cyanobacteria have been isolated and grouped in 260 families of metabolites, of which 148 (53%) belong to marine environments [2]. Their wide structural diversity is the result of their ability to integrate non-ribosomal peptide synthetases with polyketide synthases, as well as several enzymes responsible for other biosynthetic transformations, originating chemically diverse natural products such as glycolipids, macrolides, peptides (depsipeptides and lipopeptides), and polyketides, among others [3,4,5,6,7]. Recently, these metabolites have been recognized as important sources of new chemical structures with potential therapeutic uses due to their biological properties. Particularly, secondary metabolites from cyanobacteria have attracted scientific attention due to their cytotoxicity on several cancer-derived cell lines: up to 110 families of metabolites (42%), out of the 260 reported, are cytotoxic [2]. Several cyanobacteria-derived or -inspired therapeutic agents are currently under clinical investigation [8]. 

Cyanobacteria Stanier ex Cavalier-Smith is an important phylum of the kingdom Eubacteria with a single taxonomic class (Cyanophyceae Schaffner). The class Cyanophyceae contains, until now, five subclasses and 12 orders [9], but their taxonomy is under constant revision and modification. These organisms are slow-growing photosynthetic prokaryotes, and the phylum includes more than 5000 species [9], distributed in a wide range of ecosystems worldwide. Besides the members of the kingdoms Plantae (plants) and Chromista (eukaryotic microalgae), cyanobacteria are also responsible for producing atmospheric oxygen and play a substantial role in generating a strikingly different group of secondary metabolites. 

Many of the secondary metabolites from marine cyanobacteria whose cytotoxic activities have been studied and reported belong predominantly to two orders: Oscillatoriales and Synechococcales.

From the order Oscillatoriales, members of the families Oscillatoriaceae, Microcoleaceae, Gomontiellaceae, Vermifilaceae and Coleofasciculaceae, and from the order Synechococcales, members of the families Procholortrichaceae, Leptolyngbyeaceae, Prochloraceae, Merismopediaceae, and Synechococcales familia incertae sedis have been studied (Table 1 and Table 2). Cyanobacteria from the order Oscillatoriales are amongst the most productive group when referring to cytotoxic metabolites, and members of the genus *Lyngbya* produce many of them. From the cytotoxic metabolites reviewed here, only one metabolite family is common to more than one cyanobacterial order: the aplysiatoxins. 

Additionally, it is well-known that molecules from the dolastatin family have been isolated from cyanobacteria from the order Oscillatoriales. However, dolastatin 12, somamide A, and somamide B, members of the dolastatin family, have been isolated from *Lyngbya majuscula*/*Schizothrix* sp. assemblages [10,11]. These cyanobacteria belong to the orders Oscillatoriales and Synechococcales, respectively. Therefore, there is no concrete evidence of dolastatins isolated from cyanobacteria of the order Synechococcales, but there is a possibility that some molecules may be produced by cyanobacteria of this order. Therefore, further research is needed to determine the true biosynthetic origin of these molecules.

To the best of our knowledge, this is the second review focusing on anticancer properties that group cyanobacterial compounds into families of metabolites. Additionally, this review has performed a meticulous analysis focused on identifying each variant belonging to each family of metabolites and the cyanobacterial strains they come from (Appendix A presents all the information regarding these variants). This information provides a general perspective to better understand the biosynthetic and taxonomic origins of these molecules and their bioactive properties, since, until now, many of them have also been identified in other marine organisms and/or cyanobacteria from different taxonomic orders. In addition, it is expected in the near future that new cyanobacterial molecules will be identified and grouped into families of metabolites. Therefore, this review is a valuable scientific source of knowledge regarding the bioactive properties, suggested mechanisms of action, producing cyanobacterial strains, and potential structural changes that could modulate the cytotoxic properties of these new molecules. 

This review is focused on the cytotoxic, antiproliferative, and antineoplastic effects of metabolites produced by marine cyanobacteria, emphasizing the identification of each variant of the metabolite family, their chemical structures, and the mechanisms of action underlying their biological and pharmacological activities. Information on the cell lines’ origins can be found in Appendix A.

## 2. Marine Cyanobacterial Metabolites with Cytotoxic, Antiproliferative, and Antineoplastic Effects

### 2.1. Glycolipids

#### Bartolosides

Bartolosides (Figure 1) are chlorinated aromatic glycolipids recently discovered from the marine cyanobacteria *Nodosilinea* sp. and *Synechocystis salina* [18,19]. These molecules were evaluated against three cancer cell lines showing mild cytotoxicity. 

Bartoloside A exhibited IC_50_ values of 22, 40, and 23 µM against MG-63, RKO, and T-47D cells, respectively, while bartoloside E exerted IC_50_ values of 39, 40, and 22 µM, respectively. When the T-47D cell line was exposed to bartoloside I, it revealed a cytotoxic effect with an IC_50_ of 59.8 μM [19]. 

### 2.2. Macrolides

#### 2.2.1. Caylobolides

Caylobolides are macrolactones (Figure 2) isolated from the cyanobacteria *Lyngbya majuscula* and *Phormidium* spp. and constitute a family of metabolites that exhibit cytotoxic activities against several neoplastic cell lines. 

Caylobolide A exerted cytotoxicity against HCT-116 human colon tumor cells with an IC_50_ value of 9.9 µM [20]. In comparison, caylobolide B exhibited cytotoxic activity against HT-29 colorectal adenocarcinoma cells and HeLa cervical carcinoma cells with IC_50_ values of 4.5 and 12.2 µM, respectively [21]. On the other hand, nuiapolide exhibited anti-chemotactic activity against Jurkat cells with an IC_50_ value below 1.3 μM and induced cell arrest in the G2/M phase of the cell cycle [22]. Due to their related structural characteristics, the effects of nuiapolide on Jurkat cells suggest that caylobolids A and B could exert their cytotoxicity through a similar mechanism, causing various cellular responses leading to cell death.

#### 2.2.2. Swinholide-Type

Members of the swinholide-type family are macrolides with a unique dimeric 44-membered or larger lactone ring (Figure 3) that are potent cytotoxins capable of disrupting the actin cytoskeleton [23]. One dimeric macrolide binds simultaneously to two molecules of G-actin, forming a tertiary complex with the two side chains of the macrolide, thus inhibiting polymerization by the sequestration of G-actin [24,25].

The first member of this family, swinholide A, was first discovered in the mid-1980s from the marine sponge *Theonella swinhoei* by the Carmely, et al. [24] research group. Subsequently, Andrianasolo, et al. [13] isolated swinholide A and two new related molecules, ankaraholides A and B, from the cyanobacteria *Symploca* sp. and *Geitlerinema* sp., respectively.

Swinholide A exhibited cytotoxic properties against several tumor cell lines (Table 3), HT-1080 fibrosarcoma cells being the most sensitive, with an IC_50_ of 0.017 µg mL^−1^, while PC-3 lung adenocarcinoma cells were the most resistant, with an IC_50_ of 6 µg mL^−1^ [26,27].

On the other hand, ankaraholide A inhibited the proliferation of the NCI-H460, Neuro-2a, and MDA-MB-435 cell lines, with IC_50_ values of 119, 262, and 8.9 nM, respectively [13]. 

Recently, Tao, et al. [23] evaluated the cytotoxicity of nine new swinholide-type compounds, samholide A–I, isolated from the marine cyanobacterium *Phormidium* sp., which were potently active against H-460 human lung cancer cells, with IC_50_ values ranging between 170 and 910 nM.

### 2.3. Linear Peptides

#### 2.3.1. Bisebromoamides

Bisebromoamides (Figure 4) are linear peptides that impair actin dynamics and exert antiproliferative activity at the nanomolar level. These molecules have been isolated from the cyanobacterium *Lyngbya* sp. The molecule bisebromoamide has shown an IC_50_ of 40 nM against HeLa S_3_ cells and an average GI_50_ of 40 nM against a panel of 39 human cancer cell lines (JFCR39). In comparison, norbisebromoamide exerted an IC_50_ of 45 nM against HeLa S_3_ cells [28,29]. Bisebromoamide exhibited a potent inhibition of ERK in NRK cells, in addition to inducing apoptosis through the inhibition of ERK, AKT, and mTOR in 769-P and 786-O kidney cancer cells [30]. 

Bisebromoamide was the first linear peptide acting as a stabilizer of actin filaments, leading to the synthesis and evaluation of four thiazoline analogs that demonstrated nanomolar cytotoxic activity against HCT-116 human colon tumor cells through an apoptotic mechanism. These compounds exerted half-maximal effective concentration (EC_50_) values ranging between 45 and 483 nM. Among these analogs, synthetic derivatives (1) and (2) were the most cytotoxic, with an EC_50_ of 45 and 71 nM, respectively. The study revealed that the C|terminus could be altered without significantly affecting the activity. Still, that alteration of N-terminal residues or replacement of the N-terminal pivalate cap removes all cytotoxic activity. Analogs 1 and 2 (Figure 4) also caused similar morphological changes to those previously reported for bisebromoamide in HeLa cells, including a clear disruption and aggregation of the F-actin filaments and reduced cell–substrate adhesion. In addition, a reverse-phase protein array (RPPA) profiling analysis suggested a specific mode of action for these analogs, which resulted in the reduced activity of PKC and reduced expression of insulin receptor substrate 1 (IRS-1) [31].

#### 2.3.2. Carmaphycins

Carmaphycins A and B (Figure 5) are linear peptides and a new class of marine-derived epoxyketone 20S proteasome inhibitors. These molecules are isolated from the marine cyanobacterium *Symploca* sp. In their structures, these compounds contain an α,β-epoxyketone warhead derived from a leucine directly connected to a methionine sulfoxide or methionine sulfone bound to a valine and an alkyl chain terminal tail [32]. These molecules showed potent cytotoxicity against NCI-H460 human lung adenocarcinoma cells, HCT-116 colon cancer cells, and the *NCI-60 panel of tumor cell lines*, with GI_50_ values between 1 and 50 nM [33]. This mechanism of action and proven cytotoxic activity has led to the conjugation of these molecules and their analogs with antibodies to design antibody–drug conjugates (ADCs) as new anticancer drugs. Using a highly toxic derivative of carmaphycin as the warhead of ADCs could provide the necessary potency and achieve a better selectivity profile [34]. Additionally, inhibition of the proteasome, a proteolytic complex responsible for the degradation of ubiquitinated proteins, has emerged as a potent strategy in treating cancer [35].

### 2.4. Depsipeptides

#### Anaenamides

Anaenamides A and B (Figure 6) are linear depsipeptides and geometric isomers recently isolated from the green filamentous cyanobacterium *Hormoscilla* sp. (VPG16-5). These compounds contain two α-hydroxy acid residues, flanked by an alkylated salicylic fragment and an unusual α-chlorinated α,β-unsaturated (E/Z) ester, which is new to cyanobacterial natural products. Anaenamides A and B exerted moderate cytotoxicity against the cancer cell line HCT-116 with IC_50_ values of 4.5 and 8.7 µM, respectively [15]. Based on their structural features, the study revealed that the halogenated α,β-unsaturated ester moiety serves as the pharmacophore for the cytotoxic activity. 

Similarly, anaenamides C and D were isolated from a novel *Hormoscilla* sp. (VPG16-58). These compounds retain the isomeric (Z/E) chlorinated α,β-unsaturated unit reported in anaenamides A and B but differ by the presence of a primary amide instead of the methyl ester. Interestingly, anaenamides C and D showed cytotoxicity only at 100 μM against the cancer cell line HCT-116; however, both compounds activated the ARE-luciferase reporter by 15- and 17-fold, respectively, at 32 μM, without affecting the cell viability in the stable HEK293 cell line [36]. As the authors mention, these data suggest that modified pharmacophores in anaenamides lead to different targets, and cytotoxic activity can be separated from Nrf2 induction.

### 2.5. Cyclic Depsipeptides

#### 2.5.1. Apratoxins

The family of apratoxins is formed by cyclic depsipeptides with cytotoxic and antiproliferative activities at the nanomolar level (Table 4).

Apratoxins B and C exhibited cytotoxicity against KB and LoVo cells with IC_50_ values ranging from 0.73 to 21.3 nM [37]. At the same time, apratoxins D, F, G, and H and apratoxin A sulfoxide exerted cytotoxic activity against NCI-H460 cells, with IC_50_ values between 2 and 89.9 nM [38,39,40]. Similarly, apratoxins E and F showed cytotoxicity against several cancer cell lines (HT-29, HeLa, U20S, and HCT-116), with IC_50_ values ranging from 21 to 72 nM [41]. 

Apratoxin A (Figure 7a) is undoubtedly the most-studied molecule of the apratoxin family in terms of cytotoxic activity. This molecule was tested against the *NCI-60 panel of tumor cell lines* showing differential cytotoxicity. However, it was not selective for cancer cell lines derived from any particular tissue or tumor type (Table 4) [42]. This compound exerts its antiproliferative activity through: (1) the induction of G1 cell cycle arrest and an apoptotic cascade, which is initiated at least partially through the antagonism of fibroblast growth factor (FGF) signaling via STAT3 [42], and (2) downregulating numerous cancer-associated receptors and their growth factors, including multiple receptor tyrosine kinases (RTKs) [43,44]. These antitumor effects are mediated through the inhibition of the Sec61 protein translocation channel, which blocks co-translational translocation within the secretory pathway and prevents the biogenesis of secreted protein factors and integral membrane proteins, as well as the rapid proteasomal degradation of several proteins by preventing their N-glycosylation [43,45,46].

On the other hand, administering apratoxin A against *in vivo* tumor models has shown variable activity, exerting marginal antitumor activity against a mice model of early-stage colon adenocarcinoma [47] and promising therapeutic effectiveness against human HCT-116 colon cancer in SCID mice [39]. Nevertheless, in both studies, this compound proved to be toxic.

The potent cytotoxic activity against tumor cells and their novel chemical structures makes the apratoxin family an interesting depsipeptides family for potentially effective antineoplastic drug development. This interest has led to the generation of several synthetic derivatives with improved antineoplastic and antitumor activities and lower toxicity [48,49,50], apratoxin S10 (Figure 7b) being a promising one in terms of potency, stability, and synthetic accessibility [51]. This molecule inhibited angiogenesis *in vitro* through downregulation of the expression of vascular endothelial growth factor receptor 2 (VEGFR2) on endothelial cells and blocking the secretions of vascular endothelial growth factor A (VEGFA) and interleukin 6 (IL-6) from highly vascularized cancer cells (A498, HuH-7, and NCI-H727) [51]. In addition, apratoxin S10 also inhibited the proliferation of cancer cells from highly vascularized tumors through the downregulation of RTKs, including VEGFR2, epidermal growth factor receptor (EGFR), met proto-oncogene (hepatocyte growth factor receptor) (MET), insulin-like growth factor 1 receptor β (IGF1Rβ), and fibroblast growth factor receptor 4 (FGFR4) [51]. Furthermore, this compound inhibited the growth of EC68, EC46 (primary pancreatic cancer cells), and PANC-1 (exocrine pancreatic cancer cells) cells at a low nanomolar GI_50_ range and exerted antitumor effects in a pancreatic cancer patient-derived xenograft mouse model through reduced cellular proliferation without causing weight loss [52].

#### 2.5.2. Aurilides

Aurilides are members of a family of cyclic depsipeptides showing an α-hydroxy acid residue, a polyketide segment containing three or four stereogenic centers and a pentapeptide [53]. The first member of this family, the aurilide, was isolated in the 1990s from the sea hare *Dolabella auricularia.* This molecule exhibited potent cytotoxicity against the cancer cell line HeLa S_3_ with an IC_50_ of 0.011 µg mL^−1^ [54]. This family of molecules has been isolated and identified in different marine cyanobacteria of the order Oscillatoriales.

Aurilides have attracted the attention of researchers for their potent cytotoxic and antiproliferative properties. Han, et al. [55] evaluated the cytotoxicity of aurilides B and C, isolated from the marine cyanobacterium *Lyngbya majuscula*, against NCI-H460 lung carcinoma and Neuro-2a mouse neuroblastoma cells. Aurilide B showed the highest cytotoxicity, with lethal concentration (LC_50_) values of 10 and 40 nM for Neuro-2a and NCI-H460 cells, respectively, while aurilide C presented LC_50_ values of 50 and 130 nM, respectively. Considering these results, aurilide B was evaluated in the *NCI-60 panel of tumor cell lines* and showed a high level of growth inhibition on leukemia, renal, and prostate cancer cell lines, with a cell growth inhibition concentration (GI_50_) of less than 10 nM. In this sense, other family members have also exhibited interesting cytotoxic activities against neoplastic lines. Palau’amide showed potent cytotoxicity against KB cells with an IC_50_ value of 13 nM [56]; meanwhile, odoamide exhibited cytotoxicity against the HeLa S_3_ and A549 cell lines, with IC_50_ values of 26.3 nM and 4.2 nM, respectively [57,58]. Similarly, lagunamides showed potent cytotoxicity against several cell lines (P388, BJ, BJ Shp 53, PC3, SK-OV-3, HCT8, and A549) with IC_50_ values that ranged between 1.6 and 68.2 nM (Table 5) [59,60,61,62].

Several aurilide analogs have been developed to identify possible antineoplastic drug candidates, which have already been described in detail [53]. The structures of aurilide, the first member of this family, and aurilide B are shown in Figure 8.

Although aurilide, the first member of this family, has not yet been isolated from cyanobacteria, its cytotoxic mechanism of action has been the most studied in this family. This molecule selectively binds to prohibitin 1 (PHB1) in mitochondria, leading to mitochondrial fragmentation through the proteolytic processing of optic atrophy 1 (OPA1), resulting in a loss of membrane potential and the induction of apoptotic cell death [63]. These data could suggest that, due to their related structural characteristics, the mechanisms underlying the cytotoxic activities of the aurilides family could be due to the induction of apoptosis mediated by the mitochondrial function. In fact, biochemical studies using HCT8 and MCF7 cancer cells suggested that the cytotoxic effect of lagunamides A and B might act via the induction of mitochondrial-mediated apoptosis [61]. In addition, the downregulation of the *ATP1A1* gene, which encodes a catalytic α subunit of Na^+^/K^+^ ATPase, confers a significant sensitivity of cancer cells to aurilide B [64]. These data suggest that changes in *ATP1A1* gene expression could also be involved in the mechanism of cell death induced by these molecules in the cancer cell lines.

#### 2.5.3. Cocosamides

Cocosamides A and B (Figure 9) are cyclic depsipeptides that induce mild cytotoxicity against MCF7 breast cancer cells and HT-29 colon cancer cells. These compounds have demonstrated cytotoxic activity against HT-29 cells with IC_50_ values of 24 and 11 μM, respectively. MCF7 cells were slightly less susceptible to both compounds, with IC_50_ values of 30 and 39 μM, respectively [65]. 

#### 2.5.4. Coibamide A

Coibamide A (Figure 10a) is a cyclic depsipeptide with cytotoxic activities against several cell lines derived from human neoplasms. 

This molecule showed potent cytotoxicity against NCI-H460 lung cancer cells and Neuro-2a mouse neuroblastoma cells, with LC_50_ values lower than 23 nM, showing arrest in the G1 phase of the cell cycle [17]. In addition, coibamide A was evaluated against several tumor cell lines, including the *NCI-60 panel of cancer cell lines*, showing promising cytostatic and cytotoxic effects, with GI_50_ values ranging between 0.4 and 7.6 nM, being the MDA-MB-468 line being the most sensitive (Table 6) [17,44,66]. 

Hau, et al. [69] evaluated this molecule in human glioblastoma cells U87-MG and SF-295, observing a concentration- and time-dependent cytotoxicity with EC_50_ of 28.8 and 96.2 nM, respectively. According to that study, coibamide A can induce apoptosis by activating caspase-3 in SF-295 cells. Still, it can also trigger autophagy through an mTOR-independent pathway in U87-MG cells, indicating that this molecule induces biochemically different kinds of cell death, depending on the cell type.

Additionally, coibamide A has shown a variety of other anticancer activities, such as: (1) the inhibition of autophagy by blocking autophagosome–lysosome fusion due to the alteration of protein glycosylation of the lysosome membrane LAMP1 and LAMP2 in MDA-MB-231 cells [70], (2) a decreased expression of VEGFR2 and inhibition of VEGFA secretion in U87-MG and MDA-MB-231 cells [71], and (3) a decreased expression of the HER receptor family in breast cancer cells and non-small cell lung cancer cells [44]. Furthermore, this compound suppressed tumor growth in glioblastoma xenografts [71]. Coibamide A mediates its anticancer activity by inhibiting Sec61 translocon, leading to the downregulation of various proteins, including receptors and their growth factors [72].

Similar to apratoxin A, coibamide A showed toxicity against in vivo tumor models [71]. Nevertheless, due to their powerful antiproliferative effects, mechanism of action, and novel chemical structure, several analogs have recently been synthesized to develop more effective and selective antineoplastic drugs that reduce the exhibited toxicity. However, coibamide A has a well-defined conformational structure and is very sensitive to backbone modifications, resulting in analogs with significantly decreased activities [66,67]. Among these derivatives, a simplified analog, [MeAla3-MeAla6]-coibamide (3) (Figure 10), showed an inhibition of growth similar to coibamide A against MDA-MB-231, A549, and PANC-1 cancer cells, with GI_50_ values of 5.1, 7.3, and 7 nM, respectively. In addition, analog (3) inhibited tumor growth in the human MDA-MB-231 xenograft mouse model without presenting significant weight loss or side effects at the dose evaluated [66]. Furthermore, a stimuli-responsive peptide–drug conjugate (PDC) composed of a tumor-homing peptide with the cRGDyK sequence (cyclic RGD), the synthetic derivative (3), and a reduction-cleavable disulfide linker suppressed tumor growth in the A549 xenograft mouse model with negligible toxicity [68].

#### 2.5.5. Largazole

Largazole (Figure 11), originally isolated from a cyanobacterium of the genus *Symploca* (later reclassified as *Caldora penicillata* [73]), is a potent and selective class I histone deacetylase (HDAC) inhibitor. This cyclic depsipeptide is a prodrug activated by removing the octanoyl residue from the 3-hydroxy-7-mercaptohept-4-enoic acid moiety to generate the active metabolite largazole thiol, which binds to the catalytic Zn^2+^ ion of HDAC enzymes to exert its inhibitory effects [74]. 

Largazole exerts highly differential growth inhibitory activity, preferentially targeting cancer cells over non-transformed cells and showing antiproliferative activity at the nanomolar level. This activity has been tested on several neoplastic cell lines with mixed results (Table 7) [75]. 

Similarly, largazole exerted cytotoxic activity against NUT midline carcinoma cell lines [76], a panel of chemo-resistant melanoma cell lines [74], prostate cancer cell lines [77], and lung cancer cells [78], with IC_50_ values ranging between 24 and 570 nM (Table 7), showing strong increases in the acetylation of histone H3 and upregulating the expression of tumor suppressor p21 in prostate cancer cell lines PC3 and LNCaP [77] and A549 lung cancer cells [78].

Liu, et al. [82] demonstrated that exposure of HCT-116 colorectal carcinoma cells to largazole induces dose-dependent cell cycle arrest, with low concentrations (1–3.2 nM) causing G1 phase arrest, while high concentrations (10 nM) cause G2/M phase arrest, in addition to triggering apoptosis. This antiproliferative activity was correlated with the hyperacetylation of histone H3 (Lys9/14), leading to changes in the expression of genes that regulate the cell cycle, proliferation, and apoptosis. Moreover, largazole strongly stimulated histone hyperacetylation, induced apoptosis, and exhibited efficacy in inhibiting tumor growth in the human HCT-116 xenograft mouse model without toxicity. This activity was mediated by modulation of the cell cycle regulation levels, antagonism of the AKT pathway through the downregulation of IRS-1, and reduction of the epidermal growth factor receptors levels. Furthermore, Law, et al. [86] demonstrated that a combined treatment between largazole and dexamethasone reduced the invasiveness of MDA-MB-231 breast cancer cells in orthotopic xenograft tumors by inducing E-cadherin re-localization from cytoplasmic vesicles to the plasma membrane, where it facilitates cell–cell adhesion and attenuates invasion.

Additionally, largazole showed a variety of other anticancer effects, such as: (1) the inhibition of ubiquitin-activating enzyme (E1), which mediates ubiquitin conjugation to p27 and TRF1 [80], (2) sensitization of EBV^+^ lymphoma cells to the anti-herpes drug ganciclovir [87], (3) induction of the proteasomal degradation of E2F1 in lung cancer cells [78], (4) tumor growth inhibition in the A549 non-small cell lung carcinoma xenograft model [83], (5) inhibition of the oncogenic KRAS and HIF pathways in colorectal cancer cells and antiangiogenic activity in human cells and zebrafish [79], and (6) upregulation of the *Pax6* gene, which elicits the suppressed proliferation and invasion of the glioblastoma multiforme (GBM) [75].

To date, several analogs of this compound have been synthesized, and their biological activities have been evaluated [88]. A synthetic derivative termed OKI -179 was currently evaluated in a phase I clinical trial to treat advanced solid tumors. It was concluded that it has a manageable safety profile and a favorable pharmacokinetic profile and demonstrated on-target pharmacodynamic effects at tolerable doses [89]. Owing to the exceptional activity of largazole, it is expected that new synthetic analogs with higher potency and selectivity will be produced to develop anticancer drugs.

### 2.6. Cyclic Peptides and Depsipeptides

#### Dolastatins

Dolastatins represent a large family of active compounds that show a high heterogeneity, since they include linear and cyclic peptides and depsipeptides, peptides containing thiazole and oxazole heterocycles, and macrolides [90]. Although dolastatins have long been consolidated as a single family of metabolites, several authors have suggested that these compounds cannot be studied as a single group due to their significant structural variability and their marked differences in biological activities. However, the discussion in detail is beyond the scope of this review, but the reader can refer to the work by Ciavatta, et al. [90], where the origin, structure, and bioactivity of dolastatins are discussed.

These compounds were initially isolated from the sea hare *Dolabella auricularia* and, later, from various marine cyanobacteria (Table 1). Among these highly diverse compounds, dolastatin 10 and 15 (Figure 12) show the most remarkable anticancer activities in the *in vitro* and *in vivo* models.

Members of this family (dolastatins 10 and 15) are potent microtubule depolymerizers that induce cell cycle arrest in the G2/M phase and trigger apoptosis in several tumor cell lines (Table 8). Dolastatin 10 is a linear peptide originally isolated from a cyanobacterium of the genus *Symploca*, later reclassified as *Caldora penicillata* [73]. This molecule exerted cytotoxic activity against KB, LoVo, A549, and DU-145 cells, with IC_50_ values of 0.052, 0.076, 0.97, and 0.5 nM, respectively [91,92,93]. Similarly, dolastatin 10 exhibited cytotoxic activity against human lymphoma cell lines, small-cell lung cancer cells, human colon cancer cells, and ovarian tumor cells, with IC_50_ values ranging from 0.00013 to 1.8 nM [94,95,96].

Dolastatin 10 also displays *in vivo* cytotoxic activity against several models, including NCI-H446 [95] and ovarian carcinoma xenografts [96]. Owing to its mechanism of action and cytotoxic potency at the nanomolar level, dolastatin 10 entered phase I and II clinical trials for the treatment of cancer. Still, it was considered ineffective as an anticancer agent due to dose-limiting side effects, such as neuropathy, and its clinical development stopped [8,100,101]. Afterward, due to structure–activity relationship (SAR) analyses, a series of analogs named auristatins were developed; however, these also proved ineffective in clinical trials due to their toxicity. To reduce the toxicity derived from their low selectivity, auristatins were joined to a linker to facilitate their conjugation to monoclonal antibodies, leading to the generation of ADCs and the development of the Food and Drug Administration (FDA)-approved ADC brentuximab vedotin (Adcetris) and polatuzumab vedotin (Polivy). Currently, brentuximab vedotin is used to treat Hodgkin’s lymphoma and anaplastic large cell lymphoma [102,103], and polatuzumab vedotin is used in combination with bendamustine and a rituximab product for the treatment of relapsed or refractory diffuse large B-cell lymphoma [104]. These drugs consist of monomethyl auristatin E (MMAE) chemically conjugated to the antiCD30 and antiCD79b antibodies, respectively, which target tumor cells and selectively deliver MMAE to induce apoptosis. Consequently, designing and utilizing the auristatin family members within ADCs has been an active area of investigation, given its role in delivering the drug to antigen-positive tumor cells [105,106,107].

Dolastatin 15, a linear depsipeptide initially isolated from the sea hare *Dolabella auricularia* and recently from the marine cyanobacteria VPG14-73 and VPG16-8 [97] exerted cytotoxic activity against HCT-116 and A549 cancer cells with IC_50_ values of 2.2 and 0.74 nM, respectively [92,97]. This molecule also exhibited cytotoxic activity against human lymphoma, colon cancer, and ovarian tumor cells, with IC_50_ values ranging between 0.0013 and 10 nM [94,96]. Dolastatin 15 was slightly inferior to dolastatin 10 in terms of anticancer activity against the human ovarian carcinoma xenografted model [96]. Similar to dolastatin 10, dolastatin 15 and a series of analogs have been evaluated in clinical trials for the treatment of cancer, synthadotin (ILX-651) being a promising synthetic derivative for locally advanced or metastatic melanoma, since it showed low toxicity and has completed phase I and II clinical research [108,109]. Nevertheless, dolastatin 15 and other analogs have not succeeded beyond phase II clinical trials for their lack of efficacy [6,8].

Other dolastatins isolated from cyanobacteria also show cytotoxic activity at the nanomolar level (Table 8). Dolastatin 16, a cyclic depsipeptide isolated from *Lyngbya majuscula* and *Symploca* cf. *hydnoides*, exerted cytotoxic activity against NCI-H460, KM20L2, SF-295, and SK-MEL-5 cells, with IC_50_ values of 0.96, 1.2, 5.2, and 3.3 ng mL^−1^, respectively [110]. Similarly, symplostatin 1 and 3 exhibited cytotoxic activity against several cell lines (KB, LoVo, MDA-MB-435, SK-OV-3, and NCI/ADR), with IC_50_ values ranging from 0.09 to 10.3 nM [91,98,99]. Moreover, symplostatin 1 proved its potent antitumor activity against a drug-insensitive mammary tumor and a drug-insensitive colon tumor but also showed high toxicity [91].

### 2.7. Linear Lipopeptides

#### 2.7.1. Almiramides

Almiramides are lipopeptides isolated from the cyanobacteria *Lyngbya majuscula* and *Oscillatoria nigroviridis.* These compounds have exhibited toxic properties in various organisms. However, these properties also allow them to exert cytotoxicity against several tumor cell lines. In this sense, almiramides B and D (Figure 13) exhibited cytotoxic effects against the human tumor cell lines A549, MDA-MB-231, HT-29, HeLa, and PC3, with concentrations ranging from 13 to 107 µM. MDA-MB-231 cells were the most sensitive line to almiramide B, with an IC_50_ of 13 µM, whereas the HeLa cell line was the most susceptible to almiramide D, with an IC_50_ of 17 µM [111]. 

#### 2.7.2. Microcolins

Microcolins are lipopeptides isolated from the cyanobacteria *Moorea producens* and *Lyngbya polychroa* (Figure 14). 

Microcolins A–M induce significant cytotoxicity against NCI-H460 human lung cancer cells, with IC_50_ values ranging from 6 nM to 1 µM, microcolin A being the most cytotoxic molecule of this family against this cell line [112]. On the other hand, microcolins A and B and deacetylmicrocolin B inhibit HT-29 and IMR-32 cell growth, with IC_50_ values ranging between 0.28 and 14 nM [113].

#### 2.7.3. Wenchangamides

Wenchangamide A (Figure 15a) is a lipopeptide recently isolated from the marine filamentous cyanobacteria *Neolyngbya* sp. Wenchangamide A showed growth inhibitory activity against HCT-116 human colorectal cancer cells with IC_50_ values of 38 µM. This effect was related to cell cycle arrest at the G2/M phase and the induction of apoptosis. However, the same cell cycle arrest or apoptosis pattern was not observed in normal human dermal fibroblasts (NHDF), suggesting a specificity to cancer cells [14]. Wenchangamide A has a structural resemblance to minnamide A (Figure 15b), a cyanobacterial natural product of *Okeania hirsute*. However, the main structural differences are the length polypeptide core scaffold and the longer fatty acid tail. Minnamide A has an N-Me-Val–Ser–N-Me-Val moiety instead of the N-Me-Phe group present in Wenchangamide A. Similarly, minnamide A showed cytotoxic activity in HeLa with an IC_50_ value of 0.17 μM, inducing necrosis through the generation of lipid ROS [114].

### 2.8. Cyclic Lipopeptides

#### Hectochlorins

Hectochlorins are cyclic lipopeptides that have demonstrated strong abilities to promote actin polymerization, and their structure resembles the lyngbyabellins class of compounds. Hectochlorin (Figure 16a) showed cytotoxic activity against CA46, KB, and NCI-H187 cells, with IC_50_ values of 0.02, 0.86, and 1.2 μM, respectively. It also caused the arrest in the G2/M phase of the cell cycle in CA46 cells [115,116]. Hectochlorin was also tested on the NCI-60 panel of cancer cell lines and showed the highest potency against the colon, kidney, and melanoma cancer cell lines, with an average GI_50_ of 5.1 µM [115]. On the other hand, hectochlorin B (deacetylhectochlorin), originally isolated from the sea hare *Bursatella leachii* and, later, from the marine cyanobacterium *Moorea producens* [117], exerted cytotoxic activity against the KB and NCI-H187 cell lines, with IC_50_ values of 0.31 and 0.32 µM, respectively [116].

### 2.9. Peptolides

#### 2.9.1. Lyngbyabellins

Lyngbyabellins are cyclic or heteromeric depsipeptides and lipopeptides isolated from various marine cyanobacteria that display thiazole rings, hydroxy acid residues, and an acyl group with distinctive chlorination at the penultimate carbon atom [118] (Figure 17). Several lyngbyabellins have been reported to exhibit cytotoxicity against various cancer cell lines by disrupting the actin cytoskeleton [119,120]. 

Lyngbyabellins A–D exhibited moderate cytotoxicity against KB and LoVo cells, with IC_50_ values below 5.3 µM [119,121,122,123]. Similarly, lyngbyabellins E–I showed cytotoxicity against NCI-H460 human lung tumor cells and Neuro-2a mouse neuroblastoma cells, with LC_50_ values ranging between 0.2 and 4.8 µM [120]. On the other hand, Choi, et al. [118] evaluated the cytotoxicity of lyngbyabellins K–N against HCT-116 and NCI-H460 cancer cells, lyngbyabellin N being the only molecule that presented cytotoxicity with an IC_50_ value of 40.9 nM for HCT-116 and values between 0.0048 and 1.8 µM for NCI-H460. Meanwhile, lyngbyabellins A, B, H, J, and P and 27-deoxylyngbyabellin A were evaluated against several cancer cell lines (HT-29, HeLa, MCF7, and CA46) and showed cytotoxic activity with IC_50_ values that oscillated between 7.3 nM and 9 µM, 27-deoxylyngbyabellin A being the most potent molecule against the HT-29 and HeLa lines, lyngbyabellin H against MCF7 cells, and lyngbyabellin B against CA46 cells (Table 9) [115,124,125,126].

#### 2.9.2. Majusculamides 

Majusculamides are cyclopeptolides isolated from the marine cyanobacterium *Lyngbya majuscule* with important cytotoxic properties (Figure 18). 

Majusculamides C and D exhibited cytotoxic activity against several cell lines, with IC_50_ values ranging from 0.32 nM to 1.3 µM [127,128]. In addition, desmethoxymajusculamide C exhibited potent cytotoxicity against various cancer cell lines by disrupting the cytoskeletal actin microfilament networks, the human colon carcinoma cell line HCT-116 being the most sensitive, with an IC_50_ value of 20 nM (Table 10 ) [129]. 

#### 2.9.3. Patellamides

Patellamides (Figure 19), members of the cyanobactin superfamily, are cyclic octapeptides that contain oxazolines and thiazoles [130] and are produced by the marine cyanobacterium *Prochloron didemni*, the symbiont of ascidian *Lissoclinum patella* [16,130].

Patellamides display several biological activities, including cytotoxicity and the ability to reverse multidrug resistance. Patellamides A–C showed cytotoxic activity against the L1210 cell line, with IC_50_ values ranging from 2 to 3.9 µg mL^−1^. In addition, patellamide A was active against CEM acute leukemia cells, with an IC_50_ of 0.028 µg mL^−1^ [131]. On the other hand, patellamides B–D acted as resistance-modifying agents in the multidrug-resistant CEM/VLB_100_ human leukemic line towards vinblastine, colchicine, and adriamycin treatments [132,133].

### 2.10. Polyketides

#### 2.10.1. Aplysiatoxins

Aplysiatoxins embody a 45-member family of metabolites, from which eleven have shown variable cytotoxic effects, while others have even demonstrated tumor growth. 

Aplysiatoxin producers are members of the orders Oscillatoriales (*Lyngbya majuscula, Lyngbya* sp., *Oscillatoria nigroviridis*, *Oscillatoria* sp., *Moorea producens*, and *Trichodesmium erythraeum*) and Synechococcales (*Schizothrix calcicola*). Some of these molecules strongly enhance the protein kinase C (PKC) activity and elicit potent tumor-promoting actions [134,135]. However, other compounds from this family have shown cytotoxic effects against several cell lines. Notably, eleven aplysiatoxins showed cytotoxic activity against L1210 mouse leukemia cells at concentrations ranging between 4.6 and 10 µg mL^−1^ [136,137,138], oscillatoxin I (Figure 20) being the most potent molecule against this cell line, with an IC_50_ of 4.6 µg mL^−1^ [139]. 

#### 2.10.2. Caldorazole

Caldorazole (Figure 21) is a new polyketide that contains two thiazole rings and an O-methylenolpyruvamide moiety. This compound was recently isolated from the marine cyanobacterium *Caldora* sp. The molecule showed strong growth inhibitory activity against solid tumor cell lines HeLa, CaSki, and HT-1080, with IC_50_ values of 0.023, 0.068, and 0.074 μM, respectively. In addition, it exhibited cytotoxic activity against three HeLa cell lines (HeLa, HeLa S3, and HeLa S3Mer–), with IC_50_ values ranging between 0.023 and 0.048 μM. Interestingly, caldorazole was shown to inhibit energy metabolism in cancer cells by blocking mitochondrial complex I activity, leading to selective cytotoxicity against human solid tumor cell lines under glucose-restricted conditions [140].

### 2.11. Metabolites from Other Chemical Families

#### 2.11.1. Iezoside

Iezoside (Figure 22) is a novel peptide–polyketide hybrid glycoside isolated from the marine cyanobacterium *Leptochromothrix valpauliae*. According to Kurisawa, et al. [141], this molecule is the first natural product that contains a 2,3-O-dimethyl-α-l-rhamnose branch between an α,β,γ,δ-unsaturated amide group and a conjugated diene group. In addition, this compound includes a main chain composed of an odd number of carbon atoms and a β-branch methyl group in the polyketide moiety, two unusual structural features.

This compound showed potent antiproliferative activity against HeLa cells, with an IC_50_ value of 6.7 nM, cell cycle delay, and spindle-like morphological changes. Iezoside exerts its antiproliferative activity by inducing endoplasmic reticulum (ER) stress via the inhibition of sarco/endoplasmic reticulum Ca^2+^-ATPase (SERCA), a membrane protein on the ER that transports Ca^2+^ from the cytosol into the ER, which activates both the cell-cycle delay and the apoptosis-signaling pathways [141].

#### 2.11.2. Santacruzamate A 

Santacruzamate A (Figure 23) is an HDAC inhibitor isolated from the marine cyanobacterium cf. *Symploca* sp. Santacruzamate A (CAY10683) is an organooxygen and organonitrogen compound derived from a gamma amino acid. Gromek, et al. [142] depicted that this compound consists of three structural moieties: an ethyl carbamate terminus, a modified γ-aminobutyric acid (GABA) linker, and a phenethylamine cap group. 

This molecule was first believed to be a potent and selective HDAC2 inhibitor (IC_50_ 0.019 and 0.112 nM for the natural and synthetic compounds, respectively) and a cytotoxic agent against HCT-116 colon carcinoma cells and HuT-78 cutaneous T-cell lymphoma cells [143]. Nevertheless, a later study by the same research group showed that santacruzamate A inhibited all 11 HDACs at a concentration of 10 μM, with little to no inhibition at 5 μM, indicating that the IC_50_ values for these isozymes all are in the 5–10 μM range. In addition, it was not cytotoxic against HCT-116 and HuT-78 cells [142]. 

Recently, Zhou, et al. [144] showed that the individual treatment of santacruzamate A or individual therapy with tacedinaline (selective HDAC1 inhibitor) did not show cytotoxicity toward hepatocellular carcinoma cells (HepG2 and Huh7). Interestingly, a combined therapy of santacruzamate A with tacedinaline showed growth inhibition, cell cycle progression changes, and apoptosis induction in the HepG2 and Huh7 cell lines. Additionally, Zhang, et al. [145] showed that the combined treatment of santacruzamate A with imatinib exerted a synergistic effect on the inhibition of cell viability, induction of apoptosis, and cell cycle arrest in the G2/M phase in imatinib-resistant chronic myeloid leukemia (CML) cells. At the same time, it produced minimal effects on normal CD34^+^ cells. In addition, using a xenograft mouse model, the combination treatment suppressed CML proliferation *in vivo*, extending the overall survival without causing apparent weight loss or toxicity. The authors suggested that these effects occur primarily through HDAC2 inhibition, implicating the modulation of the PI3K/Akt signaling pathway. Interestingly, individual santacruzamate A treatment also showed inhibition of the viability and induction of apoptosis in imatinib-resistant CML cells (LAMA84, K562, and mononuclear cells from patients), as well as anticancer effects *in vivo*, but less pronounced than combination therapy. Furthermore, santacruzamate A reduced the tumorigenic ability of WM451 and A375 cells; increased their radiosensitivity in the subcutaneous sarcoma mouse model; and decreased the HIF-1α, Ki67, and LDHA protein levels in tumor tissues of mice [146]. Therefore, it is necessary to continue exploring the biological origin of these activities to propose their potential efficacy in cotreatments with antineoplastic drugs.

Despite these contrasts, the investigation of santacruzamate A synthetic derivatives has shown cytotoxic activity against MCF7 and HCT-116 cells, with IC_50_ values between 0.476 and 23.7 μM. [142,147]. Among these analogs, the synthetic derivative (4), a tertbutyldimethylsilyloxy (OTBS) cap group analog with an intact ethyl carbamate and GABA-derived linker groups, exhibited the most cytotoxic activity against MCF7 cells, with an IC_50_ of 13.8 µM. In contrast, no cytotoxic activity was found in noncancerous peripheral blood mononuclear cells (PBMC) [142]. On the other hand, the synthetic derivative (5) containing a cap group analog with an intact ethyl–carbamate moiety showed the most antiproliferative activity against HCT-116 cells, with an IC_50_ of 0.476 µM [147]. In addition, an HDAC inhibitory activity was not detected in these analogs, suggesting that their cytotoxic activity is exerted by an HDAC-independent mechanism.

## 3. Future Perspectives

Despite the research efforts in recent decades, cancer incidence and mortality continue to rise. According to the World Health Organization (WHO), cancer is one of the leading causes of death worldwide, with almost 10 million deaths in 2020, generating a significant public health problem, since it causes great loss of human lives and economic resources [148,149]. To face this public health problem, a wide range of experimental approaches have been proposed recently to address the challenges associated with the development and progression of cancer. Unfortunately, these efforts not only have resulted in marginal advances, but it is well-known that the drugs currently used for anticancer therapies have significant side effects in these patients, impacting their quality of life. This scenery has encouraged scientists to search for new biologically active compounds that represent promising strategies for developing therapeutic agents used in treating this disease. 

Recently, marine natural products and their derivatives have been recognized as important sources of new chemical structures with potential for the development of therapeutic agents, since, in addition to their structural diversity, they represent novel mechanisms of action for several biological processes [6,150]. One of the elements associated with the great variety of activities is the broad spectrum of secondary metabolites in various organisms. Secondary metabolites fulfill essential functions during interactions between the organism, other organisms, and their environment as defense compounds or signal molecules. However, these metabolites could also represent promising chemical strategies for disease control or eradication. Cyanobacteria have been identified as one of the most promising organisms, as they are responsible for producing a strikingly distinct group of secondary metabolites with a wide range of bioactivities. These bioactivities include antiproliferative, antimicrobial, antifungal, antituberculosis, antisuppressive, and anti-inflammatory, as the potential anticancer effects worthy of emphasis [7,32]. Some of the cyanobacterial compounds and their analogs have been successfully used in clinical trials against cancer [8,151], and to date, more than 1600 molecules of cyanobacterial origin have been isolated. Most of these compounds (148 families of metabolites, 53% of families of metabolites) are produced by marine cyanobacteria [2]. Nonetheless, aplysiatoxins, aurilides, dolastatins, patellamides, and swinholide-type compounds have also been isolated from various marine non-cyanobacterial organisms, such as mollusks [93,152], ascidians [132], and sponges [153,154], which makes the true biosynthetic origin of this class of natural products confusing and ambiguous. Although there is much debate in the literature, several studies have suggested that these compounds are produced by cyanobacteria that act as symbionts with these organisms or due to the ability of marine organisms to sequester, store, and use secondary metabolites from cyanobacteria as a chemical defense mechanism against predators or even to compete for nutrients and space [16,154,155,156].

Despite the large number of compounds reported in the literature, the search for natural marine products with pharmacological properties still presents some critical challenges, such as the difficult access to various regions of the ocean, the insufficient quantities of isolated compounds, and the lack of available information on the genomes of several producer organisms. The difficulties are likely to be overtaken using innovative technologies that can assist in obtaining, isolating, evaluating, and identifying compounds and their producer organisms. These technologies include extreme environment sampling, hyphenated techniques for structural determination at the nanomole scale, nanomole NMR techniques, computational chemistry and database, genome mining, molecular networking, and others [157,158,159]. These approaches will allow the search for new bioactive marine compounds more efficiently and will lead to the identification of new molecules, improving the response to existing antitumor treatments and/or representing an important source of new chemical structures with importance for the development of antineoplastic drugs.

In this review, an effort has been made to present cytotoxic, antineoplastic, and antiproliferative compounds of marine cyanobacterial origin, the producing species, and the reported mechanisms of action underlying their pharmacological activities. 

It is essential to emphasize that the metabolites-producing cyanobacteria were referenced in the present review as described in the original articles. The genera *Moorea*, *Okeania*, *Limnoraphis*, and *Microseira*, previously grouped in the genus *Lyngbya,* have been recently reassigned based on molecular and phylogenetic analyses, and according to Demay, et al. [2], some previously morphologically identified marine strains, such as *Lyngbya majuscula* and *Lyngbya sordida*, have been renamed as *Moorea producens*. In addition, some *Lyngbya bouillonii* strains have been renamed *Moorea bouillonii* [2]. Moreover, marine cyanobacteria specimens traditionally identified as *Lyngbya penicilliformis* and *Symploca* spp. have been assigned to the new genus *Caldora*, designating *C. penicillata* as a generitype [73]. 

Some of the families of metabolites shown here have exhibited substantial cytotoxicity and antineoplastic activities against several cancer cell lines at the nanomolar level. Among these compounds, apratoxins, aurilides, bisebromoamides, carmaphycins, dolastatins, coibamide A, iezoside, and largazole can be mentioned. The mechanisms involved in their cytotoxicity are diverse. Still, some families of metabolites exert similar mechanisms (Figure 24), such as bisebromoamides, hectochlorins, lyngbyabellins, majusculamides, and swinholide-type, which impair the actin cytoskeleton; largazole and santacruzamate A, which are HDAC inhibitors; and apratoxins and coibamide A, which inhibit the trimeric Sec61 translocon, causing diverse cellular responses, depending on the cell type. Furthermore, some families show other interesting mechanisms, such as dolastatins, causing the depolymerization of microtubules, carmaphycins inhibiting the 20S proteasome, aurilides binding to PHB1 leading to mitochondrial fragmentation, caldorazole blocking mitochondrial complex I activity, iezoside inhibiting SERCA, and patellamides preventing multidrug resistance. Although the mechanisms of action of many marine cyanobacterial compounds are unknown, the ability to induce apoptosis has been observed. This effect has been related to several apoptotic markers, such as cell cycle arrest, promotion and activation of the caspase cascade, and modification of the levels of specific proapoptotic proteins [160,161]. 

In this review are also included some underreported families of metabolites, such as anaenamides, bartolosides, caylobolides, cocosamides, almiramides, microcolins, and aplysiatoxins. In this sense, the lack of interest in these families could be due to their lower potency (µM range) compared to other families of metabolites (nM range). However, compounds of these or other underexplored families of metabolites could have unknown cytotoxic mechanisms of action or could exhibit greater profiles of selectivity towards cancer cells compared to normal cells. Unfortunately, these parameters have scarcely been explored in most compounds of cyanobacterial origin. Moreover, most neoplasms have a marked resistance to chemotherapeutic agents, even in *in vitro* models, such as the well-documented case of temozolomide (TMZ) in cells derived from glioblastomas. A concentration in the range of 500–1000 µM is required to reach an IC_50_ [162] on these cells, not to mention that these chemotherapeutic agents, currently used in cancer therapy, have significant negative impacts on normal cells. Therefore, the fact that these little-explored families may have new cytotoxic mechanisms of action, mechanisms to avoid drug resistance, and/or selectivity profiles suitable for the development of new anticancer agents cannot be ignored, which is further incentive to continue exploring in greater detail the bioactive properties of the under-explored families of cyanobacteria.

On the other hand, it has been proven that apratoxins and coibamide A can be toxic against normal cells; however, their high levels of cytotoxicity against tumor cells make them interesting lead compounds. Therefore, structural modifications are required to generate analogs with differential cytotoxicity profiles that lead to the development of more potent and selective anticancer drugs. In fact, several studies have evaluated the synthetic analogs of these compounds, as mentioned in their respective sections, finding that some show greater efficacy than the original molecule, in addition to presenting less toxicity *in vivo*. Moreover, due to its novel mechanism of action, which downregulates the receptors and growth factor ligands through the inhibition of the Sec61 protein translocation channel, it is expected that these compounds may be helpful in the treatment of neoplasms stimulated mainly by secreted growth factors, such as colorectal, hepatocellular, and pancreatic carcinomas [50], and/or for the treatment of highly vascularized tumors, such as renal, hepatocellular, and neuroendocrine carcinomas [51].

Furthermore, cancer cells show a great diversity of genetic and epigenetic changes and the capacity to regulate certain protein levels to promote cell survival, proliferation, and/or to inhibit normal cell death mechanisms [163,164]. Recently, the expression levels of SERCA isoforms have been an active area for cancer research due to their role in evading apoptosis and activating metabolic stress, recognizing their regulation as a promising drug target for cancer treatments [165]. Similarly, due to the reprogramming of energetic metabolism in cancer cells, many tumors are primarily glycolysis-dependent, attracting substantial attention to glycolysis inhibition as a new central issue for drug development against specific tumors [164]. In this sense, it is expected that largazole, iezoside, and caldorazole may be useful for developing drugs that can assist in treating neoplasms with these particular hallmarks. 

In addition to the anticancer effects in preclinical models, several marine cyanobacterial-derived compounds have emerged as templates for developing new anticancer drugs. Examples of these are the FDA-approved brentuximab vedotin (Adcetris) and polatuzumab vedotin (Polivy), drugs inspired by the compound dolastatin 10. It should be noted that several synthetic derivatives of the different molecules from the aforementioned families of metabolites are in clinical studies to treat several types of cancer [8,105], and many more are expected to enter clinical trials due to new approaches to designing ADCs and PDCs. 

Recent advances in molecularly targeted therapies against cancer represent revolutionary approaches to treating these patients. Bioactive antineoplastic molecules isolated from cyanobacteria constitute an optimistic scenario for the discovery and design of the so-called “undruggable” anticancer agents. Even when intensive research for the development and clinical testing of these innovative molecular agents is needed, the potential of these targeted treatments might constitute a hallmark of unprecedented progress for cancer patients with few therapeutic options. One of the main goals in cancer research is to propose personalized or combined therapies, for which further research to elucidate the cellular mechanisms or death pathways related to the antineoplastic molecules from marine cyanobacterial origin is essential. 

Finally, there are other major challenges in this area—for instance, (i) the isolation and identification of new organisms; (ii) having access to these organisms for research purposes; and (iii) establishing large-scale culture systems to obtain adequate biomass quantities for extracting, purifying, characterizing, and evaluating new molecules.

## 4. Conclusions

Nowadays, cancer is one of the most significant challenges in public health. Secondary metabolites produced by marine cyanobacteria constitute a promising source of compounds with potential anticancer effects due to their great structural diversity and novel cytotoxic mechanisms of action. These compounds are outstanding molecules for developing innovative antineoplastic drugs with fewer side effects in patients. Most of the available information on metabolites isolated from marine cyanobacteria is limited to depicting their cytotoxic activity in preclinical models without ultimately elucidating the underlying cytotoxic mechanisms. It is necessary then to highlight that many of these molecules have been pharmacologically underexplored and could be promising candidates waiting for more intense efforts to reveal their antineoplastic potential. Therefore, it is expected that, with the continuous progress in their study, the elucidation of the mechanisms of action involved in their cytotoxic, antiproliferative, and anticancer effects, as well as biosafety, will be answered, and some of these molecules will soon be proposed as leading compounds of interest for developing promising chemotherapeutic agents for clinical treatments. 

## Figures and Tables

**Figure 1 molecules-27-04814-f001:**
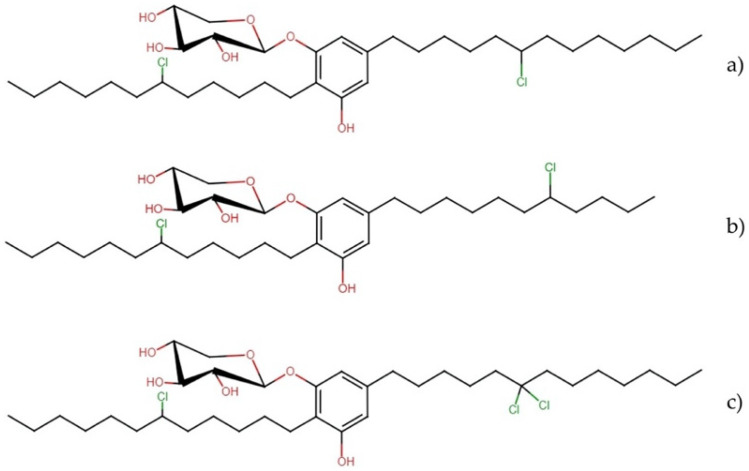
Chemical structures of bartolosides: (**a**) bartoloside A, (**b**) bartoloside E, and (**c**) bartoloside I.

**Figure 2 molecules-27-04814-f002:**
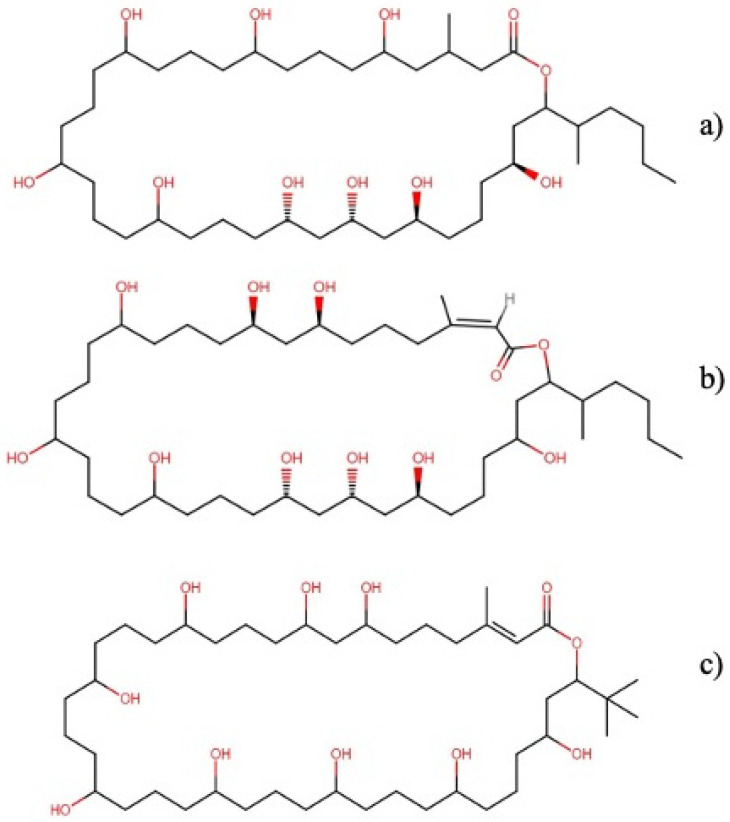
Chemical structures of (**a**) caylobolide A, (**b**) caylobolide B, and (**c**) nuiapolide.

**Figure 3 molecules-27-04814-f003:**
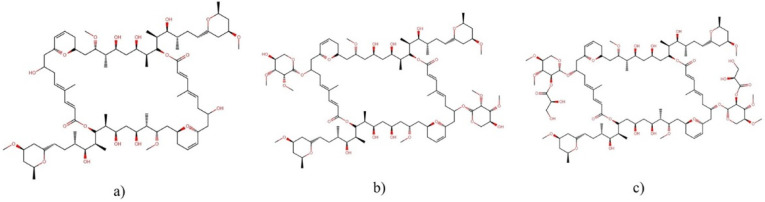
Chemical structures of (**a**) swinholide A, (**b**) ankaraholide A, and (**c**) samholide A.

**Figure 4 molecules-27-04814-f004:**
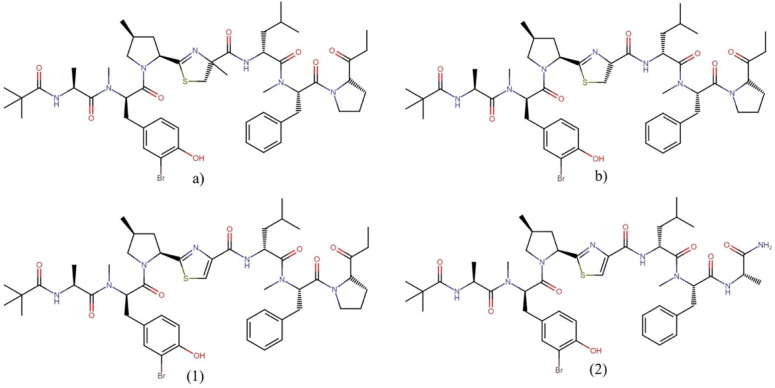
Chemical structures of (**a**) bisebromoamide, (**b**) norbisebromoamide, and their derivatives (**1**) and (**2**).

**Figure 5 molecules-27-04814-f005:**
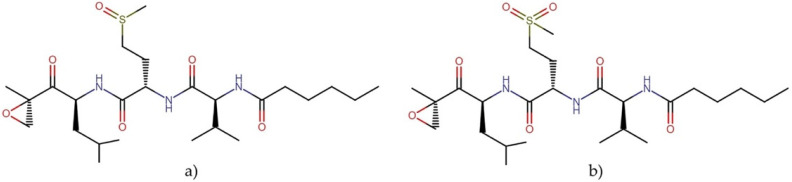
Chemical structures of (**a**) carmaphycin A and (**b**) carmaphycin B.

**Figure 6 molecules-27-04814-f006:**
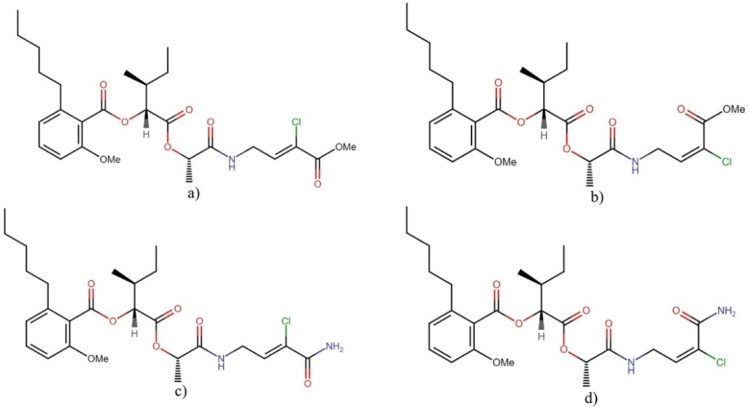
Chemical structures of anaenamides: (**a**) anaenamide A, (**b**) anaenamide B, (**c**) anaenamide C, and (**d**) anaenamide D.

**Figure 7 molecules-27-04814-f007:**
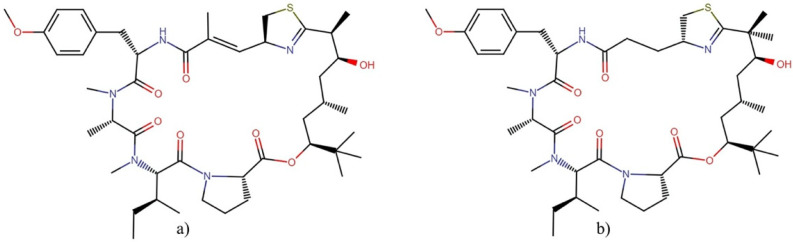
Chemical structures of (**a**) apratoxin A and (**b**) apratoxin S10. All molecules from this family are available in Appendix A.

**Figure 8 molecules-27-04814-f008:**
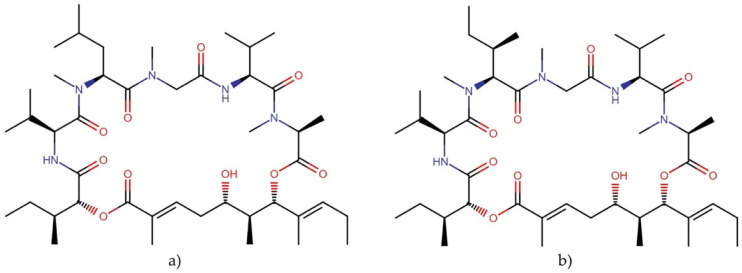
Chemical structures of (**a**) aurilide and (**b**) aurilide B. All molecules from this family are shown in Appendix A.

**Figure 9 molecules-27-04814-f009:**
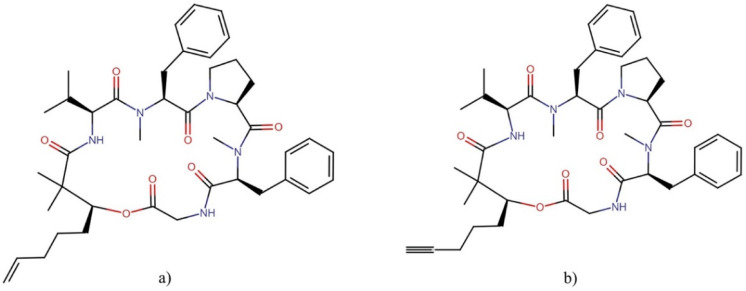
Chemical structures of (**a**) cocosamide A and (**b**) cocosamide B.

**Figure 10 molecules-27-04814-f010:**
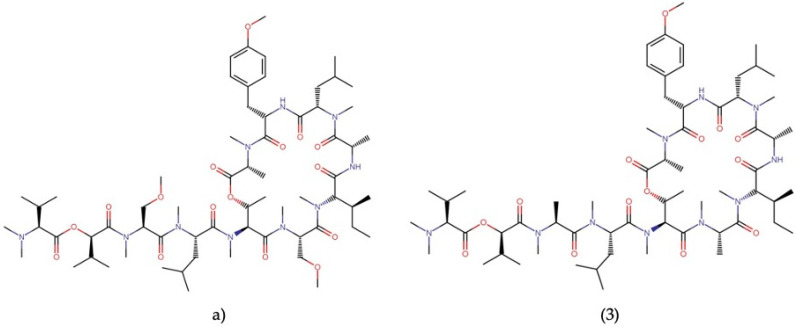
Chemical structures of (**a**) coibamide A and its derivative (**3**).

**Figure 11 molecules-27-04814-f011:**
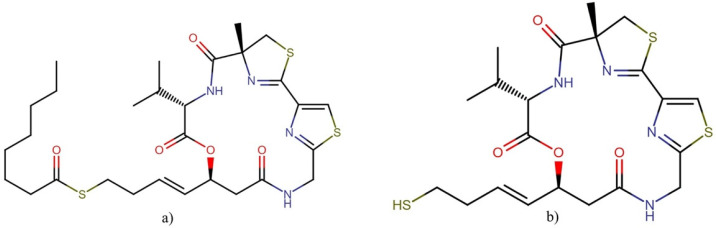
Chemical structures of (**a**) largazole and (**b**) largazole thiol.

**Figure 12 molecules-27-04814-f012:**
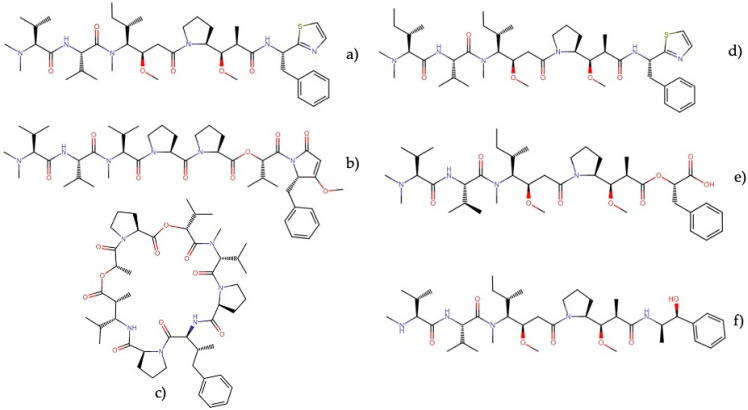
Chemical structures of representative dolastatins and their derivatives: (**a**) dolastatin 10, (**b**) dolastatin 15, (**c**) dolastatin 16, (**d**) symplostatin 1, (**e**) symplostatin 3, and (**f**) MMAE.

**Figure 13 molecules-27-04814-f013:**
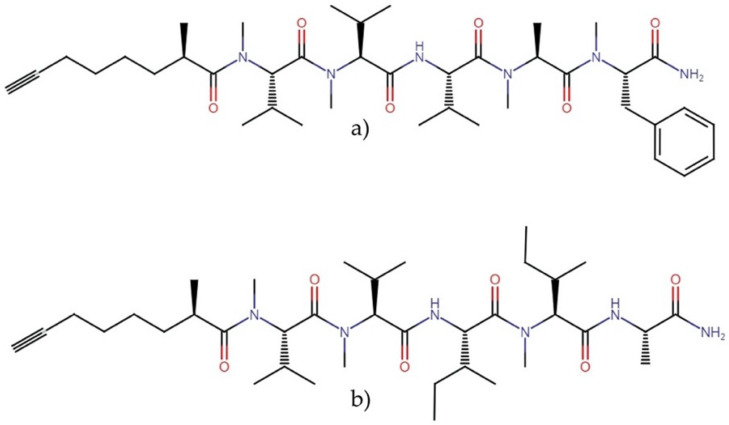
Chemical structures of (**a**) almiramide B and (**b**) almiramide D.

**Figure 14 molecules-27-04814-f014:**
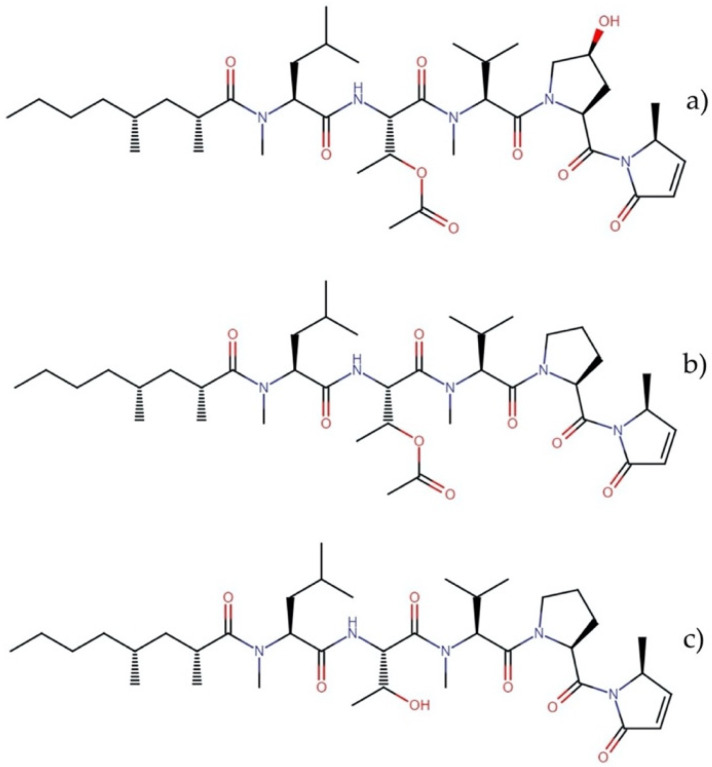
Chemical structures of (**a**) microcolin A, (**b**) microcolin B, and (**c**) deacetylmicrocolin B.

**Figure 15 molecules-27-04814-f015:**
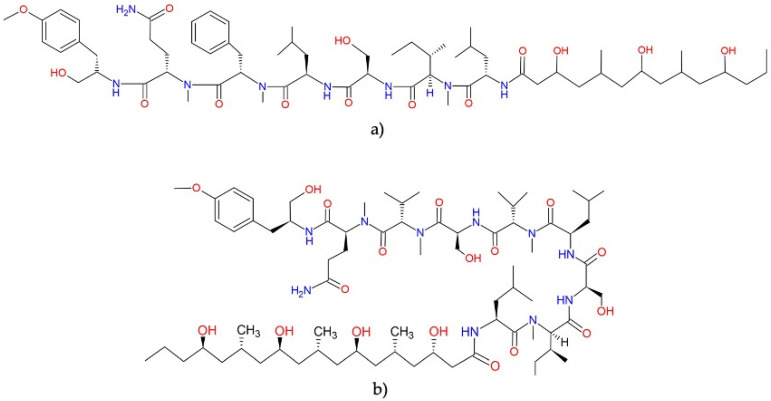
Chemical structures of (**a**) wenchangamide A and (**b**) minnamide A.

**Figure 16 molecules-27-04814-f016:**
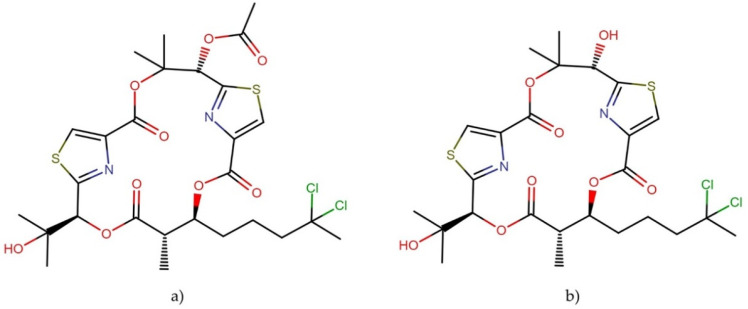
Chemical structures of (**a**) hectochlorin and (**b**) hectochlorin B.

**Figure 17 molecules-27-04814-f017:**
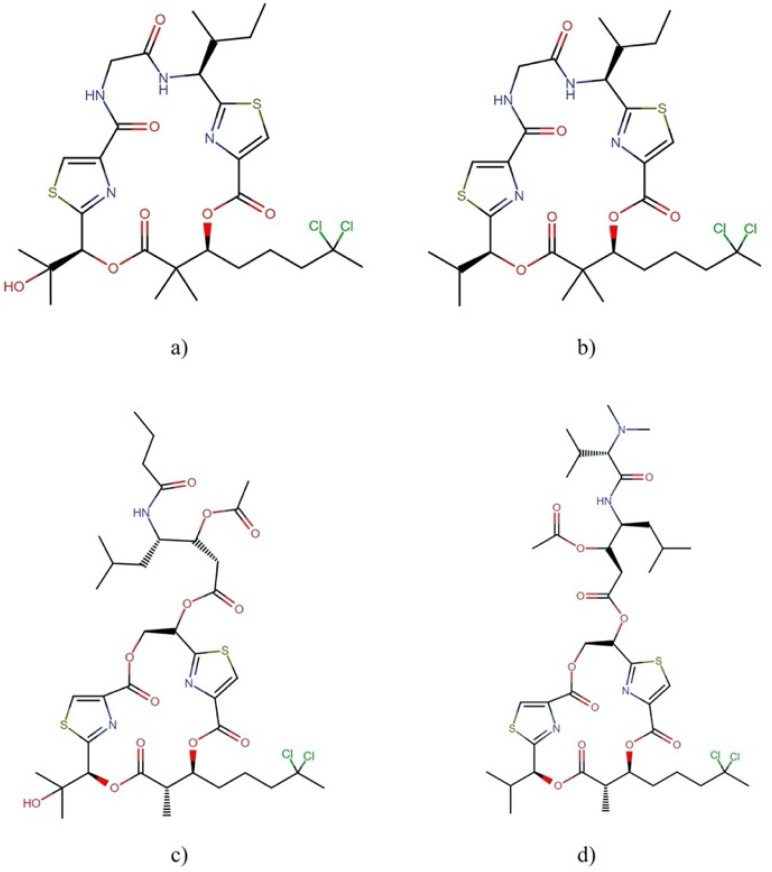
Chemical structures of the most representative lyngbyabellins: (**a**) lyngbyabellin A, (**b**) 27-deoxylyngbyabellin A, (**c**) lyngbyabellin E, and (**d**) lyngbyabellin N. All molecules from this family are shown in Appendix A.

**Figure 18 molecules-27-04814-f018:**
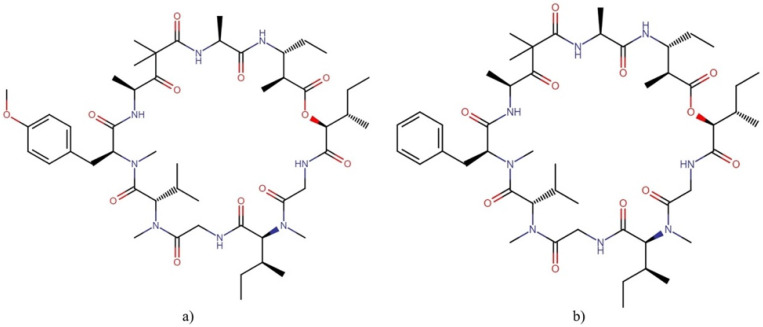
Chemical structures of (**a**) majusculamide C and (**b**) desmethoxymajusculamide C.

**Figure 19 molecules-27-04814-f019:**
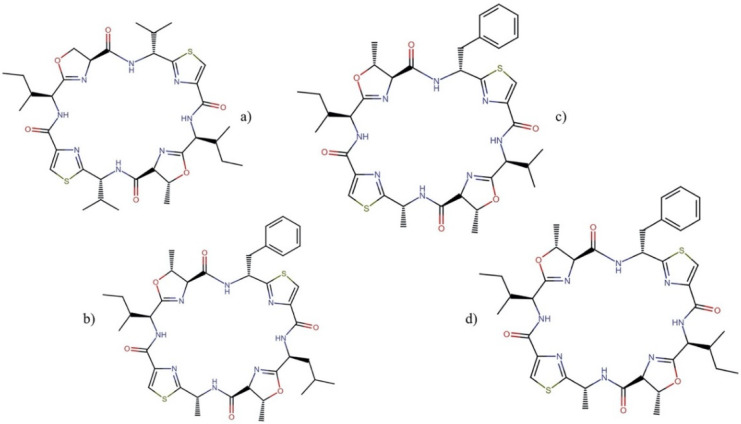
Chemical structures of the most representative patellamides: (**a**) patellamide A, (**b**) patellamide B, (**c**) patellamide C, and (**d**) patellamide D.

**Figure 20 molecules-27-04814-f020:**
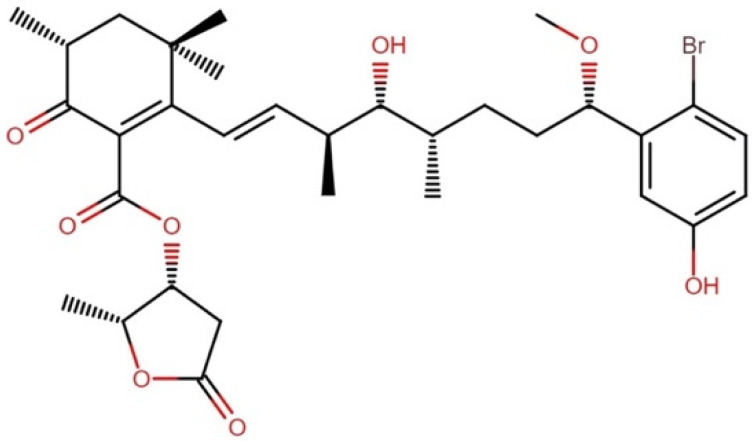
Chemical structure of oscillatoxin I.

**Figure 21 molecules-27-04814-f021:**
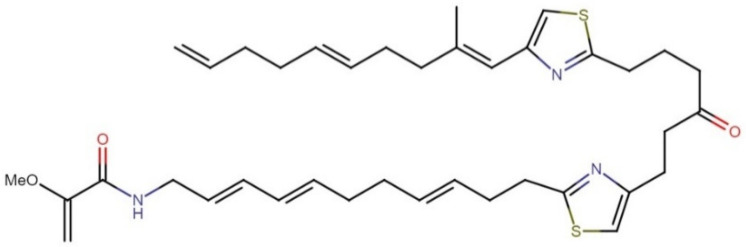
Chemical structure of caldorazole.

**Figure 22 molecules-27-04814-f022:**
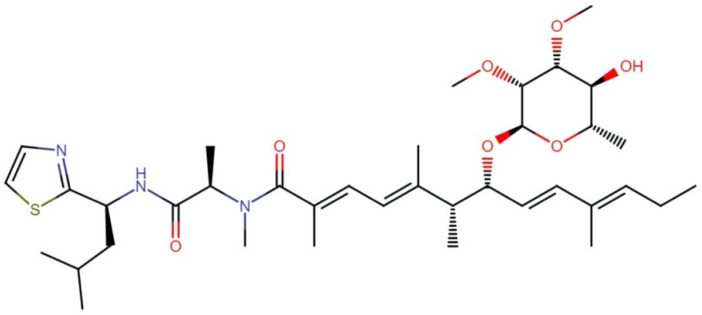
Chemical structure of iezoside.

**Figure 23 molecules-27-04814-f023:**
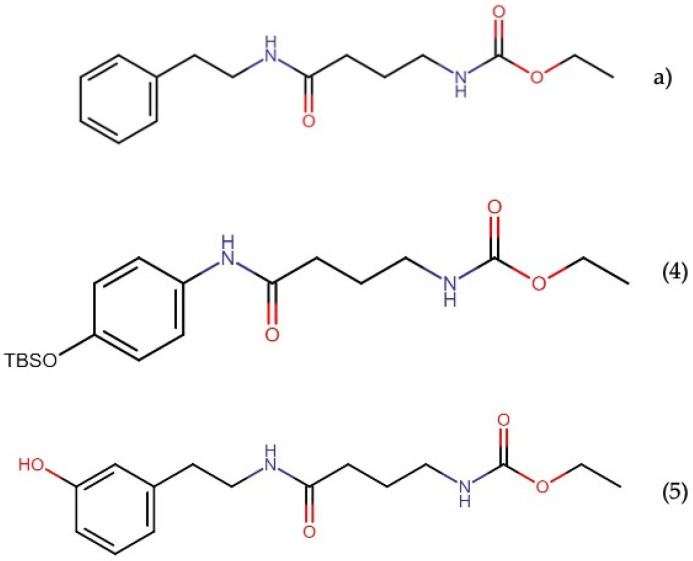
Chemical structures of (**a**) santacruzamate A and derivatives (**4**) and (**5**).

**Figure 24 molecules-27-04814-f024:**
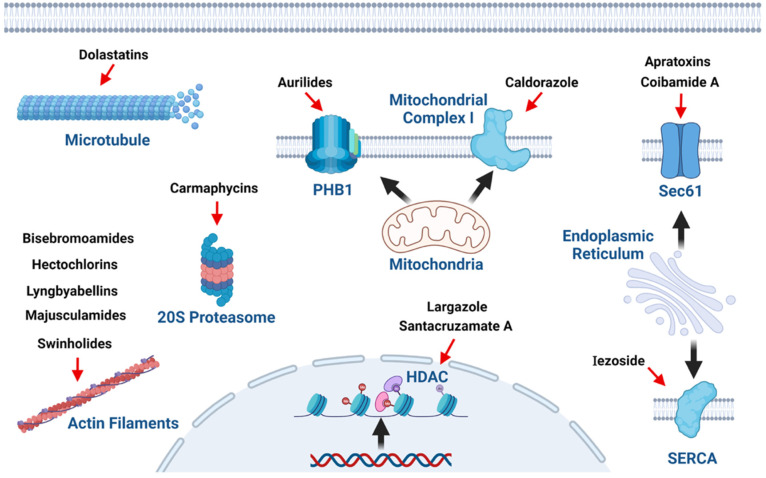
Suggested molecular targets of metabolite families isolated from marine cyanobacteria related to anticancer effects (Created by Robles-Bañuelos et al. with BioRender.com). PHB1, prohibitin 1; HDAC, histone deacetylase; and SERCA, sarco/endoplasmic reticulum Ca^2+^-ATPase.

**Table 1 molecules-27-04814-t001:** Metabolites produced by members of orders Oscillatoriales and Synechococcales.

Order	Family	Genera	Metabolites	Selected References
Oscillatoriales	Oscillatoriaceae	*Lyngbya*, *Oscillatoria*, *Moorea*, *Okeania*, *Phormidium* *Neolyngbya*	Almiramides, anaenamides, **aplysiatoxins**, apratoxins, aurilides, bisebromoamides, carmaphycins, caldorazole, caylobolides, cocosamides, dolastatins, hectochlorins, iezoside, largazole, lyngbyabellins, majusculamides, microcolins, santacruzamate A, swinholide-type, wenchangamides	[2,12,13,14,15]
	Coleofasciculaceae	*Geitlerinema*		
	Gomontiellaceae	*Hormoscilla*		
	Microcoleaceae	*Trichodesmium*, *Symploca*, *Caldora*		
	Vermifilaceae	*Leptochromothrix*		
Synechococcales	Procholortrichaceae	*Nodosilinea*	**Aplysiatoxins**, bartolosides, coibamide A, dolastatins, patellamides	[10,16,17,18]
	Leptolyngbyeaceae	*Leptolingbya*		
	Prochloraceae	*Prochloron*		
	Merismopediaceae	*Synechocystis*		
	Schizotrichaceae	*Schizothrix*		

Metabolites reported in more than one cyanobacterial order are in bold. Most of the compounds have been reported only from one genus.

**Table 2 molecules-27-04814-t002:** Chemical families of cyanobacterial metabolites.

Chemical Family		Metabolites
Glycolipids		Bartolosides
Macrolides		Caylobolides Swinholide-type compounds
Peptides		Bisebromoamides Carmaphycins Dolastatin 10 *
	Subfamily: Depsipeptides Linear depsipeptides Cyclic depsipeptides	Anaenamides Apratoxins Aurilides Cocosamides Coibamide A Largazole Dolastatins 15 and 16 *
	Subfamily: Lipopeptides Linear lipopeptides Cyclic lipopeptides	Almiramides Microcolins Wenchangamides Hectochlorins
	Peptolide ^†^	Lyngbyabellins Majusculamides Patellamides ^#^
Polyketides		Aplysiatoxins Caldorazole
Metabolites from other chemical families		Iezoside Santacruzamate A

* These are described together in a single section. ^†^ Peptolide: cyclic or heteromeric depsipeptide. ^#^ Members of the cyanobactin superfamily.

**Table 3 molecules-27-04814-t003:** Cytotoxic activity of the swinholide family on cancer cell lines. IC_50_ values.

Cell Line	Molecule			References
	Swinholide A	Ankaraholide A	Samholide A–I	
L1210	0.03			[13,23,26,27]
KB	0.04			
SW-480	0.07			
KATO-III	0.05			
HT-1080	0.017			
T-24	0.046			
PC-3	6.0			
PC-8	0.12			
PC-9	0.13			
PC-10	0.11			
PC-13	0.10			
QG-56	0.04			
Daudi	0.036			
NCI-H460		119 *	170–910 *	
Neuro-2a		262 *		
MDA-MB-435		8.9 *		

All IC_50_ values are reported in µg mL^−1^, except for *, which are reported in nM.

**Table 4 molecules-27-04814-t004:** Apratoxin analog activities on cancer cell lines.

Apratoxin Analog	Cell Line (IC_50_/* GI_50_ Values in nM)										Selected References
	KB	LoVo	NCI-H460	HT-29	HeLa	U20S	HCT-116	NCI-60 ^1^	Neuro-2a	Panel HCCL ^2^	
A	0.52	0.36	2.5	1.4	10	10	1.21	1–3 *	1000	4.9–41 *	[37,38,39,40,41,42]
A SO *			89.9								
B	21.3	10.8									
C	1	0.73									
D			2.6								
E				21	72	59					
F			2				36.7				
G			14								
H			3.4								

^1^ Most affected cells from the NCI-60 panel: HT-29, RPMI-8226, SR, and LOX IMVI. ^2^ Panel of human cancer cell lines: BxPC-3, A549, HuH-7, MKN74, U87-MG, SK-OV-3, HEC-6, and 786-O. A SO *: apratoxin A sulfoxide.

**Table 5 molecules-27-04814-t005:** Aurilide analog activities on cancer cell lines.

Molecule	Cell Line (IC_50_ / LC_50_ Values in nM)											References
	HeLa S_3_	P388	BJ	BJ Shp 53	PC3	SK-OV-3	HCT8	Neuro-2a	NCI-H460	KB	A549	
Aur B								10 *	40 *			[55,56,57,58,59,60,61]
Aur C								50 *	130 *			
Palau										13		
Odo	26.3										4.2	
Lag A		6.4	20.2	58.8	2.5	3.8	1.6				2.9	
Lag B		20.5					5.2					
Lag C		24.4			2.6	4.5	2.1				2.4	
Lag D											7.1	
Lag ’											68.2	

Aur = Aurilide, Palau = Palau’amide, Odo = Odoamide, and Lag = Lagunamide. * LC_50_

**Table 6 molecules-27-04814-t006:** Coibamide A activity on cancer cell lines.

Tested Cell Lines	LC_50_/EC_50_/GI_50_ Values (nM)	Cellular Effects	References
NCI-H460 and	LC_50_ ≤ 23	Arrest in the G1 phase of the cell cycle; mTOR-independent autophagy; apoptosis; decreased protein expression due to the inhibition of Sec9	[17,44,67,68]
Neuro-2a			
MDA-MB-231	GI_50_ = 1–7.4		
LOX IMVI	GI_50_ = 7.4		
HL-60	GI_50_ = 7.4		
SNB-75	GI_50_ = 7.6		
U87-MG	EC_50_ = 28.8		
SF-295	EC_50_ = 96.2		
A549	GI_50_ = 5.4		
PANC-1	GI_50_ = 3.1		
MDA-MB-436	EC_50_ = 0.9		
MDA-MB-468	EC_50_ = 0.4		
HS578T	EC_50_ = 4.0		
BT474	EC_50_ = 0.8		

**Table 7 molecules-27-04814-t007:** Largazole biological activity on cancer cell lines.

Tested Cell Lines	GI_50_ or IC_50_ Values (nM)	Cellular Effects	References
MDA-MB-231	GI_50_ = 7.7	Histone deacetylases inhibitor; apoptosis; modulation of the levels of cell cycle regulators; antagonism of the AKT, KRAS, and HIF pathways; reduction of the epidermal growth factor receptors levels; inhibition of ubiquitin-activating enzyme (E1); proteasomal degradation of E2F1; antiangiogenic activity; upregulation of the *Pax6* gene	[74,75,76,77,78,79,80,81,82,83,84,85]
U2OS	GI_50_ = 55		
HT-29	GI_50_ = 12		
IMR-32	GI_50_ = 16		
HCT-116	GI_50_ = 80		
A549	GI_50_ = 320		
SK-OV-3	IC_50_ = 250		
HeLa	IC_50_ = 170		
Eca-109	IC_50_ = 100		
Bel 7402	IC_50_ = 170		
U937	IC_50_ = 20		
797	IC_50_ = 24		
10326	IC_50_ = 25		
PC3	IC_50_ ≤ 500		
LNCap	IC_50_ ≤ 500		
Panel of melanoma cell lines	IC_50_ = 45–315		
NCI-H1975	IC_50_ = 83		
NCI-H460	IC_50_ = 120		
GLC-82	IC_50_ = 190		
L78	IC_50_ = 570		
SPC-A1	IC_50_= 140		
95D	IC_50_ = 420		
NCI-H466	IC_50_ = 520		
SW620	IC_50_ = 26.5		
MiaPaCa	IC_50_ = 206.4		
SH-SY5Y	IC_50_ = 102		
SF-268	IC_50_ = 62		
SF-295	IC_50_ = 68		

**Table 8 molecules-27-04814-t008:** Dolastatins and symplostatins activities on the cancer cell lines.

Cell Line (IC_50_ Values in nM)	Dolastatin (D) or Symplostatin (Sy)				References
	D10	D15	Sy1	Sy3	
KB	0.052		0.15–0.20	3.9	[91,92,93,94,95,96,97,98,99]
LoVo	0.076		0.34–0.50	10.3	
A549	0.97	0.74			
MDA-MB-435			0.15		
SK-OV-3			0.09		
NCI/ADR			2.9		
DB, HT, RL, SR	0.00013–0.13	0.0013 to 0.13			
H82, H446, H69, H510	0.032–0.184				
BE, HT-29, MAWI, SW480, SW620	0.018–0.16	0.12–2.24			
A2780, CHI, 41M, HX/62	0.046–1.8	0.061–10			
L1210	0.4	3			
DU-145	0.5				
HCT-116		2.2			

**Table 9 molecules-27-04814-t009:** Lyngbyabellin analog activities on cancer cell lines.

Lyngbyabellin	Cell Line									References
	HT-29	HeLa	KB	LoVo	NCI-H460	Neuro-2a	HCT-116	MCF7	CA46	
A	0.047 *	0.022 *	0.03 **	0.5 **						[115,118,119,120,122,123,124,125,126]
27-Deoxy A	0.012 *	0.0073 *						0.31 *		
B	1.1 *	0.71 *	0.1 **	0.83 **					0.1 *	
C			2.1	5.3						
D			0.1							
E					0.4 ^‡^	1.2 ^‡^				
F					1 ^‡^	1.8 ^‡^				
G					2.2 ^‡^	4.8 ^‡^				
H					0.2 ^‡^	1.4 ^‡^		0.07 *		
I					1 ^‡^	0.7 ^‡^				
J	0.054 *	0.041 *								
N					0.0048–1.8 *		0.0409 *			
P								9 *		

* IC_50_ Values in µM; ** IC50 Values in µg mL^−1^; ^‡^ LC_50_ Values in µM.

**Table 10 molecules-27-04814-t010:** Majusculamide analog activities on cancer cell lines.

Cell Line	Molecules			References
	Majusculamide C	Majusculamide D	Desmethoxy C	
PANC-1		0.32		[127,128,129]
U251N		36.8		
HepG2		1396		
NCI-H125		147		
P388	3.3 *			
SF-295	33 *			
HCT-116			20	
H-460			63	
MDA-MB-435			220	

IC_50_ values in nM. * GI_50_ values in ng mL^−1^.

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
