# Peer review of "Marine Cyanobacteria as Sources of Lead Anticancer Compounds: A Review of Families of Metabolites with Cytotoxic, Antiproliferative, and Antineoplastic Effects"

_molecules, 2022, doi:10.3390/molecules27154814_

Round 1

Reviewer 1 Report

General comments

1. In this review paper, the authors reported the example of each variant of the metabolite family with representative chemical structures. However, the reason behind the selection of representative compounds is not so apparent and well written, and the proper justification will be more interesting for readers. For example, the representative compounds are more potent than other compounds from similar class of metabolites.

2. Comprehensive reports are available in the previous literature on anticancer compounds from marine cyanobacteria (ex. Ref. 6). From this viewpoint, I can not see the point of reporting a similar group of compounds unless they add unique values compared to the previous ones. Authors should clarify precisely how this kind of cumulative approach is different in independently providing the information on marine cyanobacteria leading anticancer compounds.

3. The significance and implications of this review are not clearly stated. The reports are redundant in many ways, as those can be found in the other published reviews in a similar approach. Authors may specify their intention based on the gaps in the earlier literature they are developing for this review article.

4. Overall, what are core new pieces of information this literature would provide to the scientific community? Please clarify. 

Specific comments

1.      In line 108, the IC50 value of santacruzamate A was mentioned as 0.112 nM. In the reference [19] https://pubs.acs.org/doi/full/10.1021/np400198r, the IC50 value of natural santacruzamate A was reported as 0.119 nM, and the IC50 value of synthetic compound was 0.112 nM.

2.      Santacruzamate A is not a carboxylic acid.

3.      In lines 143-150, the authors mentioned the cytotoxic activities of caylobolide A, caylobolide B, nuiapolide, and the mechanism of cytotoxicity of nuiapolide. If we can see the structure of nuiapolide in the same figure with caylobolide A and B it will be more helpful for understanding as those might exhibit similar mechanisms for cytotoxicity.

4.      In Figure 3, the structure of caylobolide B is drawn as the original paper where it indicated the E-configuration between C-2/C-3, but it was corrected later as Z-configuration based on the NOE analysis has been presented in Salvador-Reyes et al., J. Nat. Prod201578, 1957–1962.

5.      The structure of swinholides in Figure 4 is too small.

6.      In the example of peptides: linear, cyclic, depsipeptides, and lipopeptides (line 177), it will be easier for readers if the order is maintained like all examples of lipopeptides together, then examples of depsipeptides and so on. Each class should be grouped as a section.

7.      Dolastatins should be treated as dolastatins 10 and 15. Dolsatatins include dolastatins 13 and 16 which are structurally different from 10 and 15. In terms of activity, 10 and 15 are clearly different from 13 and 16. This affects to many pages. Dolastatin cannot be treated as one group.

8.      Most of compounds have been reported from one genera. This information should be written in Table 1.

Author Response

Dear Editor,

Thank you and the reviewers for your generous suggestions and comments, which are very important for improving the quality of our article. We have carefully gone through our manuscript following the reviewers' recommendations. The "Response to Reviewers" is below. We responded to each point raised, either by replying to the reviewers or by adding new information to the manuscript. We have also uploaded a complete, editable electronic (Word) copy of the revised manuscript, new figures and information in the Supplementary Material. In addition, other figures have been modified and improved. We sincerely hope this revised version of the manuscript is suitable for publication.

Reviewer 1

General comments

  1. In this review paper, the authors reported the example of each variant of the metabolite family with representative chemical structures. However, the reason behind the selection of representative compounds is not so apparent and well written, and the proper justification will be more interesting for readers. For example, the representative compounds are more potent than other compounds from similar class of metabolites.
  2. Comprehensive reports are available in the previous literature on anticancer compounds from marine cyanobacteria (ex. Ref. 6). From this viewpoint, I can not see the point of reporting a similar group of compounds unless they add unique values compared to the previous ones. Authors should clarify precisely how this kind of cumulative approach is different in independently providing the information on marine cyanobacteria leading anticancer compounds.
  3. The significance and implications of this review are not clearly stated. The reports are redundant in many ways, as those can be found in the other published reviews in a similar approach. Authors may specify their intention based on the gaps in the earlier literature they are developing for this review article.
  4. Overall, what are core new pieces of information this literature would provide to the scientific community? Please clarify. 

Reply/Revised:

Thank you very much for these invaluable comments. We agree that the relevance of this review to the field of marine cyanobacteria with anticancer potential was not sufficiently emphasized in our previous manuscript. In this new version, we corrected and highlighted this importance.

Since the four main comments of reviewer 1 are generally related, we respectfully decided to provide more detailed information to expand and clarify the importance and implications this review could bring to the scientific community. These comments are:  

  • Currently, the reviews on compounds isolated from cyanobacteria only group molecules by chemical class or biological activity. Our review is the second report grouping cyanobacterial compounds into families of metabolites. Here, we have included more families than previous reviews and have provided detailed information, highlighting the cytotoxic, anti-proliferative, and antineoplastic activities of these molecules.

  • In addition to proposing a new approach by grouping these cyanobacterial molecules into metabolite families, we also performed a meticulous analysis focused on identifying each metabolite family variant, whether they are cytotoxic molecules, and the cyanobacterial strains from which they originate (all information regarding these variants is presented in Supplementary Table 1). Furthermore, we have taken great care to present only those variants that exhibit cytotoxic, antiproliferative, and/or antineoplastic activity. We intend to provide an overview to better understand the biosynthetic and taxonomic origin of these molecules, as well as their bioactive properties, since, to date, many of these molecules have also been identified in other marine organisms and/or cyanobacteria from different taxonomic orders. Finally, it is expected that, in the near future, new cyanobacterial molecules will be identified and grouped in metabolite families. Therefore, this review represents a valuable scientific source of knowledge on the bioactive properties, suggested mechanisms of action, producing cyanobacterial strains, and potential structural changes that could modulate the properties of these molecules.

  • We have included some tables on the cytotoxic properties of these metabolite families, which could help to highlight the potency among metabolites from the same families and among families. Similarly, a significant effort has been made to illustrate most of the structures of the cytotoxic molecules of the metabolite families mentioned here (particularly those with several variants), which can be found in the main text or the supplementary material, based on their primary cytotoxic function.

  • In addition, we have included some under-reported metabolite families, such as bartolosides, cocosamides, almiramides, microcolins, and aplysiatoxins, as well as under-reported variants of some metabolite families since most reviews on cyanobacterial compounds focus on the most potent molecules in terms of cytotoxicity. In this sense, we consider that the lack of interest in these metabolite families due to their lower potency (µM range) compared to other metabolite families (nM range) could generate a gap, as compounds from these metabolite families, or other poorly explored families, could have new cytotoxic mechanisms of action, such as epigenetic regulators or potentiators or synergistic (to name a few), which could present higher selectivity profiles towards cancer cells compared to normal cells. Unfortunately, these targets have been scarcely explored in most of the compounds isolated from cyanobacteria, which is a further incentive to continue studying in more detail the bioactive properties of these little-explored families.

  • We have added four new families of very recently isolated metabolites with significant cytotoxic properties and exciting mechanisms of action.

Based on the Reviewer's valuable comments, we have added several lines in the Introduction and Future Perspectives sections that address these issues, and we have also rewritten some paragraphs to include and highlight the new information reinforcing the relevance of this manuscript to the scientific community (Lines 91-104 and 803-819 [Lines without Track Changes]). Also, we have expanded several sections and added new chemical structures. In addition, we have tried to give a more applied approach to the potential benefits some cyanobacterial molecules could represent against several neoplasms based on their hallmarks and the mechanisms of action of the compounds mentioned here (Lines 820-843). We hope these included changes will satisfy the criterion of the Reviewer. 

Specific comments

  1. In line 108, the IC50value of santacruzamate A was mentioned as 0.112 nM. In the reference [19] https://pubs.acs.org/doi/full/10.1021/np400198r, the IC50 value of natural santacruzamate A was reported as 0.119 nM, and the IC50 value of synthetic compound was 0.112 nM.

Reply/Revised: Following this valuable recommendation, the statement has been corrected (Lines 673 and 674).

  1. Santacruzamate A is not a carboxylic acid.

Reply/Revised: We apologize for the confusion. Based on the Reviewer’s comment, santacruzamate A has been modified and regrouped. In addition, we have added a statement to reinforce the structure of the compound corrected (Table 2, Lines 669 and 670).

  1. In lines 143-150, the authors mentioned the cytotoxic activities of caylobolide A, caylobolide B, nuiapolide, and the mechanism of cytotoxicity of nuiapolide. If we can see the structure of nuiapolide in the same figure with caylobolide A and B it will be more helpful for understanding as those might exhibit similar mechanisms for cytotoxicity.

Reply/Revised: According to the Reviewer’s recommendation, the nuiaploide structure has been added to Figure 2 (Line 146).

  1. In Figure 3, the structure of caylobolide B is drawn as the original paper where it indicated the E-configuration between C-2/C-3, but it was corrected later as Z-configuration based on the NOE analysis has been presented in Salvador-Reyes et al.,  Nat. Prod201578, 1957–1962.

 Reply/Revised: Thank you for the valuable comment. We have corrected the structure of caylobolide B (Now Figure 2, line 146).

  1. The structure of swinholides in Figure 4 is too small.

Reply/Revised: Following this valuable recommendation, the structure of swinholides has been enlarged (Now Figure 3, Line 158).

  1. In the example of peptides: linear, cyclic, depsipeptides, and lipopeptides(line 177), it will be easier for readers if the order is maintained like all examples of lipopeptides together, then examples of depsipeptides and so on. Each class should be grouped as a section.

Reply/Revised: Thanks for the valuable comment. Following this recommendation, the families have been reordered (Table 2 and the rest of the document).

  1. Dolastatins should be treated as dolastatins 10 and 15. Dolsatatins include dolastatins 13 and 16, which are structurally different from 10 and 15. In terms of activity, 10 and 15 are clearly different from 13 and 16. This affects to many pages. Dolastatin cannot be treated as one group.

Reply/Revised: Thank you for this valuable comment. Certainly, Dolastatins is a structurally diverse family of compounds. Furthermore, many of the compounds of this family have been isolated exclusively from the mollusk Dolabella auricularia (dolastatins 11, 13, 14, 17, 18, etc.); many have been identified in D. auricularia and in cyanobacteria (dolastatins 3, 10, 12, 15, and 16); and some exclusively in cyanobacteria (Supplementary Table 1). Interestingly, dolastatins have long been considered a single family of metabolites (Demay et al., 2019; Ciavatta et al., 2017), but we agree with the reviewer, based on the current data from several authors, which strongly suggests that dolastatins cannot be treated anymore as a single group due to their significant structural variability and their marked difference in biological activities (Lines 446-449). Even when this interesting issue is beyond the scope of this review, we have been cautious to correctly identify and highlight the cytotoxic activity of those molecules that have been identified in marine cyanobacteria, especially dolastatin 10 and 15 (Line 453-455). Furthermore, based on this valuable recommendation, we have added several lines in the dolastatins section to clarify and reinforce this significant point of discussion according to current literature, adding some references so that the interested reader can investigate these issues in more detail (Lines 449-451). We hope this brief discussion will satisfy the criterion of the Reviewer.   

  • Demay, J.; Bernard, C.; Reinhardt, A.; Marie, B., Natural products from cyanobacteria: Focus on beneficial activities. Mar Drugs 2019, 17, (6), 320.
  • Ciavatta ML, Lefranc F, Carbone M, Mollo E, Gavagnin M, Betancourt T, Dasari R, Kornienko A, Kiss R. Marine Mollusk-Derived Agents with Antiproliferative Activity as Promising Anti-cancer Agents to Overcome Chemotherapy Resistance. Med Res Rev. 2017 Jul;37(4):702-801.

  1. Most compounds have been reported from one genera. This information should be written in Table

Reply/Revised: This information has been included in the table 1 as suggested by the reviewer (Line 106).

Reviewer 2 Report

The review by the authors covers selected marine cyanobacterial compounds having cytotoxic, anti-proliferative and antineoplastic properties.  However, some of the molecules presented in the review could be omitted as they are either do not have significant cytotoxic activities (e.g. bartolosides, almiramides, and cocosamides) or are not relevant (e.g. aplysiatoxins).  It would be better for the review to have more focus by only discussing molecules with significant anticancer activity and also to include recent discoveries:

1. Anaenamides

2. caldorazole

3. Iezoside

4. Somocystinamide A

In Table 2, should cyanobactins be included? For instance, patellamides are classified as cyanobactins.

The impact of the review paper could be raised by including the structures of synthetic analogs based on the natural products. For example, the authors can include chemical structures of synthetic analogs based on santacruzamate A.

Author Response

Dear Editor,

Thank you and the reviewers for your generous suggestions and comments, which are very important for improving the quality of our article. We have carefully gone through our manuscript following the reviewers' recommendations. The "Response to Reviewers" is below. We responded to each point raised, either by replying to the reviewers or by adding new information to the manuscript. We have also uploaded a complete, editable electronic (Word) copy of the revised manuscript, new figures and information in the Supplementary Material. In addition, other figures have been modified and improved. We sincerely hope this revised version of the manuscript is suitable for publication.

Reviewer 2

The review by the authors covers selected marine cyanobacterial compounds having cytotoxic, anti-proliferative and antineoplastic properties.  However, some of the molecules presented in the review could be omitted as they are either do not have significant cytotoxic activities (e.g. bartolosides, almiramides, and cocosamides) or are not relevant (e.g. aplysiatoxins).  It would be better for the review to have more focus by only discussing molecules with significant anticancer activity and also to include recent discoveries:

  1. Anaenamides
  2. caldorazole
  3. Iezoside
  4. Somocystinamide A

Reply/Revised: Thank you very much for these invaluable comments. This is a very important observation. We believe that the lack of interest in these families could be due to their lower potency (µM range) compared to other families of metabolites (nM range). However, some compounds from these or other under-explored families could have new cytotoxic mechanisms of action or could present greater profiles of selectivity towards cancer cells compared to normal cells. Unfortunately, these parameters have been scarcely explored in most compounds isolated from cyanobacteria. These little-explored families may reveal new cytotoxic mechanisms of action, such as epigenetic regulators, mechanisms to avoid drug resistance, and/or high selectivity profiles suitable for developing new anticancer agents, and they should not be ignored. These little families could represent a further incentive to continue exploring in greater detail the bioactive properties of these families of cyanobacteria. In this sense, we have added some lines in the Future Perspectives section (Lines 803-819) addressing this observation to emphasize this interesting point of discussion. We hope this brief discussion will satisfy the criterion of the Reviewer. 

Additionally, according to the Reviewer’s recommendation, we have added four new metabolite families (anaenamides, Lines 216-236; wenchangamides, Lines 535-551; caldorazole, Lines 638-649, and iezoside, Lines 651-666), three of the four suggested, and we have illustrated their mechanisms of action in Figure 24 (Line 800).

In Table 2, should cyanobactins be included? For instance, patellamides are classified as cyanobactins.

Reply/Revised: Thanks for this valuable observation. Cyanobactins is a superfamily of metabolite families that include small cyclic peptides containing heterocyclic amino acids or derivatives of isoprenoid amino acids. However, here we refer to  "patellamides" as a family of metabolites belonging to the chemical class of peptides, more specifically peptolides, since we denote mainly in Table 2 the chemical class of the families of metabolites. However, following this valuable recommendation, a small statement has been added to Table 2 to highlight the relevance of this observation (Line 111).

The impact of the review paper could be raised by including the structures of synthetic analogs based on the natural products. For example, the authors can include chemical structures of synthetic analogs based on santacruzamate A.

Reply/Revised: Following this valuable recommendation, we have added the most representative structures of the synthetic analogs mentioned in this review:

-Derivatives (1) and (2) (Lines 196 and 197) in Bisebromoamides section.

-Apratoxin S10 (Lines 255 and 256) in Apratoxin section.

-Derivative (3) (Line 349) in Coibamide A section.

-MMAE (Lines 458 and 459) in Dolastatins section.

-Derivatives (4) and (5) (Line 713) in Santacruzamate A section.

In addition, we have expanded the sections on analogs to describe and highlight their importance in the search and development of more potent and selective anticancer drugs (Bisebromoamides, Apratoxins, Coibamide A, and Santacruzamate A sections).

Round 2

Reviewer 1 Report

I found this revised manuscript much improved compared to the earlier version. I appreciate the authors' incorporation of almost all changes recommended by another reviewer and me. The main content of summarizing the metabolites is now more streamlined. Overall I think the paper is now acceptable for publication in Molecules. 

Author Response

The co-authors appreciate the Reviewer's comments.

Reviewer 2 Report

The amide bond drawing for derivative 4 (in Figure 23) needs to be fixed.

Author Response

The co-authors really appreciate the care and time the Reviewer has dedicated to this article. 

Following the reviewer's recommendation, Figure 23 has been corrected and changed in the manuscript, and uploaded in tiff format.
